# Low wnt/β-catenin signaling determines leaky vessels in the subfornical organ and affects water homeostasis in mice

Fabienne Benz[1], Viraya Wichitnaowarat[1], Martin Lehmann[1], Raoul FV Germano[2], Diana Mihova[1], Jadranka Macas[1], Ralf H Adams[3], M Mark Taketo[4], Karl-Heinz Plate[1,5,6,7,8], Sylvaine Guérit[1], Benoit Vanhollebeke[2,9], Stefan Liebner[1,5,6]*

[1]Institute of Neurology (Edinger Institute), University Hospital, Goethe University Frankfurt, Frankfurt am Main, Germany; [2]Laboratory of Neurovascular Signaling, Department of Molecular Biology, ULB Neuroscience Institute, Université libre de Bruxelles, Bruxelles, Belgium; [3]Department of Tissue Morphogenesis, Max-Planck-Institute for Molecular Biomedicine, University of Münster, Faculty of Medicine, Münster, Germany; [4]Division of Experimental Therapeutics, Graduate School of Medicine, Kyoto University, Kyoto, Japan; [5]Excellence Cluster Cardio-Pulmonary systems (ECCPS), Partner site Frankfurt, Frankfurt, Germany; [6]German Cancer Consortium (DKTK), Partner Site Frankfurt/Mainz, Frankfurt, Germany; [7]German Center for Cardiovascular Research (DZHK), Partner site Frankfurt/Mainz, Frankfurt, Germany; [8]German Cancer Research Center (DKFZ), Heidelberg, Germany; [9]Walloon Excellence in Life Sciences and Biotechnology (WELBIO), Wallonia, Belgium

*For correspondence:
stefan.liebner@kgu.de

Competing interests: The authors declare that no competing interests exist.

**Abstract** The circumventricular organs (CVOs) in the central nervous system (CNS) lack a vascular blood-brain barrier (BBB), creating communication sites for sensory or secretory neurons, involved in body homeostasis. Wnt/β-catenin signaling is essential for BBB development and maintenance in endothelial cells (ECs) in most CNS vessels. Here we show that in mouse development, as well as in adult mouse and zebrafish, CVO ECs rendered Wnt-reporter negative, suggesting low level pathway activity. Characterization of the subfornical organ (SFO) vasculature revealed heterogenous claudin-5 (Cldn5) and Plvap/Meca32 expression indicative for tight and leaky vessels, respectively. Dominant, EC-specific β-catenin transcription in mice, converted phenotypically leaky into BBB-like vessels, by augmenting Cldn5[+]vessels, stabilizing junctions and by reducing Plvap/Meca32[+] and fenestrated vessels, resulting in decreased tracer permeability. Endothelial tightening augmented neuronal activity in the SFO of water restricted mice. Hence, regulating the SFO vessel barrier may influence neuronal function in the context of water homeostasis.

DOI: https://doi.org/10.7554/eLife.43818.001

## Introduction

In vertebrates, the endothelial blood-brain barrier (BBB) is crucial for providing a permissive microenvironment for neuronal function. During developmental brain vascularization, blood vessels undergo Wnt/β-catenin signaling, driven by Wnt7a/7b that is required for angiogenesis as well as for BBB formation (*Daneman et al., 2009*; *Stenman et al., 2008*; *Liebner et al., 2008*). In the adult, the Wnt pathway remains instrumental to maintain BBB function in endothelial cells (ECs) of the central

**eLife digest** Infections and diseases in the brain and spine can be very damaging and debilitating. Indeed, the central nervous system also needs a carefully controlled biochemical environment to survive. As such, all animals with a backbone have barriers and defenses to protect and preserve this key system. One of these is the blood-brain barrier, a physical barrier between the brain and the outside world. Where most blood vessels allow relatively free exchange of chemicals between the blood and surrounding cells, the blood-brain barrier controls what can move between the bloodstream and the brain.

Yet, there are gaps in the blood-brain barrier, specifically within structures in the brain called the circumventricular organs. These leaky vessels allow the brain cells in these regions to monitor the blood and respond to changes, for example, by triggering sensations such as hunger, thirst or nausea. It is not clear what stops the blood-brain barrier from forming in these regions and what effect the presence of a barrier would have on the brains activity, or the health and behavior of the animal.

Benz et al. have now used mice and zebrafish to examine the development and structure of the blood-brain barrier. The investigation revealed that the signals that induce the blood-brain barrier throughout the brain are absent in the circumventricular organs of both species. Next, by artificially activating a protein involved in cell-cell interactions in mice, Benz et al. created blood-brain barrier-like structures in circumventricular organs by converting the leaky vessels into tight ones. This change meant that the brain cells in these regions did not respond properly to water deprivation, which potentially may have affected the regulation of thirst in these mice.

Understanding the blood-brain barrier could have a variety of impacts on how we treat diseases in the central nervous system. This includes stroke, brain tumors and Alzheimers disease. These findings could particularly help scientists to better understand conditions that affect basic needs like thirst and hunger.

DOI: https://doi.org/10.7554/eLife.43818.002

nervous system (CNS) (*Zhou et al., 2014*). Herein activation of β-catenin/TCF signaling can be induced by two flavors of the canonical Wnt pathway mediated by the ligands Wnt7a/7b and the non-Wnt-related norrin disease protein (Ndp), binding to the receptor complexes frizzled-4/Lrp5/6/Gpr124/Reck and frizzled-4/Lrp5/6/Tspan12, respectively (*Junge et al., 2009*; *Chang et al., 2017*; *Cullen et al., 2011*; *Posokhova et al., 2015*; *Kuhnert et al., 2010*; *Wang et al., 2014*; *Cho et al., 2017*; *Vanhollebeke et al., 2015*; *Eubelen et al., 2018*).

Although a strict control of the exchange between blood and the CNS tissue by the endothelial BBB is realized in most parts of the CNS, some areas of the brain and the ciliary body of the eye are exceptions to this rule, providing a physiologically highly relevant door to the CNS. The circumventricular organs (CVOs) are a number of small midline structures found in all vertebrate brains, located around the third and fourth ventricle. CVOs have a rich capillary plexus, which physiologically lacks BBB properties (*Ganong, 2000*; *Ufnal and Skrzypecki, 2014*; *Langlet et al., 2013*; *Benarroch, 2011*). These characteristics are regarded as important sites of communication between brain and blood. Based on their function, CVOs are commonly classified into secretory and sensory organs. The median eminence, the neurohypophysis (posterior pituitary, PP), the pineal gland (PI) and the subcommissural organ (SCO) belong to the secretory group. The vascular organ of the lamina terminalis (*organum vasculosum of the lamina terminalis*, OVLT), the subfornical organ (SFO) and the area postrema (AP) are considered as sensory organs (*Ufnal and Skrzypecki, 2014*).

The leaky vessels of the CVOs evidently have different morphological and structural characteristics from those of typical BBB vessels, lacking a cellular organization as neuro-vascular unit (NVU), without distinctive astrocytic endfeet and ECs with numerous fenestrations and vesicles (*Morita et al., 2016*). Interestingly, in previous reports neither the tight junction proteins claudin-5 (Cldn5), occludin (Ocln) and zonula occludens 1 (Tjp1/ZO-1) (*Mullier et al., 2010*; *Maolood and Meister, 2010*; *Norsted et al., 2008*; *Sisó et al., 2010*), nor the transporter proteins glucose transporter 1 (Glut-1) and transferrin receptor (*Norsted et al., 2008*; *Maolood and Meister, 2009*), showed a BBB-like staining in leaky CVO ECs. In line with these vascular features, Morita et al.

showed that 10 kDa dextran accumulates in the perivascular space between the inner and outer basement membranes, whereas smaller tracers up to 3 kDa dextran diffuse into the parenchyma (*Morita et al., 2016*). Not only the endothelium, but also the perivascular space in CVOs has specific properties, being enlarged and filled by collagen fibers, fibroblasts, astrocytic processes and axons (*Morita and Miyata, 2012*).

Beside the leaky vasculature of the CVOs, free diffusion of substances into the brain parenchyma is prohibited by tanycytes, specialized cells of the ependymal lining. Tanycytes contain long processes, unlike typical ependymal cells, which project to the parenchyma of the CVOs making contact to the fenestrated vascular wall of the CVOs (*Langlet et al., 2013*; *Benarroch, 2011*; *Mullier et al., 2010*).

The detailed function of all CVOs has not intensely been explored in the past, but more recently, the SFO together with the OVLT, the median preoptic nucleus (MnPO) and the PP were identified as a functional circuit, regulating drinking behavior and water homeostasis of the organism (*Zimmerman et al., 2016*; *Oka et al., 2015*; *Augustine et al., 2018*). The SFO is a tiny organ located underneath the fornix at the foramen of Monro, protruding into the third ventricle at the meeting point with the lateral ventricles. The dense vascular network in the SFO is similarly organized in different vertebrate species. It presents with heterogeneous vessel phenotypes and can therefore be divided into two zones. Whereas the outer shell contains more BBB expressing vessels, the majority within the ventromedial core is fenestrated with a wide perivascular space. In general, the vascular density is four to five times higher than in other brain regions with tortuous vessels, exhibiting a high blood volume and slow perfusion rate, thereby contributing to high permeability rates (*Sisó et al., 2010*; *Duvernoy and Risold, 2007*; *Fry et al., 2007*; *Bouchaud et al., 1989*).

Although a cell type-specific expression analysis by single cell sequencing, as it has been performed for the brain parenchyma (*Vanlandewijck et al., 2018*), has not been conducted yet, neurons and astrocytes of sensory CVOs (SFO, OVLT and AP) were shown to express specific receptors and ion channels. Those permit them to detect several blood–derived molecules such as salts, hormones, lipids and toxins and convey this information to other parts of the brain, involved in controlling autonomic and peripheral functions (*Benarroch, 2011*; *Sisó et al., 2010*). Evidence supports that 25–60% of CVO neurons respond to signals in the circulatory system and a single neuron may respond to multiple signals such as osmolarity and angiotensin II. Most sensory CVOs play a role in the control of blood pressure, fluid and sodium balance, cardiovascular regulation, feeding and energy homeostasis and immunomodulation (*Ufnal and Skrzypecki, 2014*; *Benarroch, 2011*; *Sisó et al., 2010*; *Morita and Miyata, 2012*; *Smith and Ferguson, 2012*).

Given the fact that endothelial Wnt/β-catenin signaling is necessary for BBB development and maintenance, the question remains if the pathway is instrumental in the establishment of vascular heterogeneity in the CNS. Hence, we addressed the question if Wnt/β-catenin signaling is operational in CVO vessels during development and if local regulation of β-catenin signaling is involved in establishing a leaky vascular phenotype in CVOs. Finally, we asked if dominant activation of β-catenin in ECs can overwrite the leaky vessel fate thereby affecting CVO function.

Here we show by investigating CVOs during BAT-gal reporter mouse development that at any embryonic stage analyzed, starting from E13.5, when the first SFO primordium could be identified, to P21, no activation of β-catenin signaling in CVO vessels could be detected. Focusing on the SFO as a crucial CVO in the regulation of water homeostasis, we show that blood vessels in the caudal portion were mainly leaky evidenced by Plvap/Meca32 staining, whereas capillaries in the rostral portion of the organ were tighter. Interestingly, within the vessel continuity, individual ECs might be Plvap/Meca32$^+$//Cldn5$^-$ followed by a Plvap/Meca32$^-$//Cldn5$^+$ EC, suggesting a locally confined regulation of barrier properties. Dominant, genetic activation of β-catenin signaling (gain-of-function, GOF) in ECs resulted in tightening of CVO blood vessels, evidenced by the switch from a Plvap/Meca32$^+$//Cldn5$^-$ to a Plvap/Meca32$^-$//Cldn5$^+$ vascular phenotype. Vessel tightening was accompanied by a significant reduction in endothelial fenestrations, that likely contributed to the reduction of transcellular permeability evidenced by decreased tracer leakage. Interestingly, endothelial tightening did not coincide with the formation of astrocytic endfeet towards a BBB-like NVU. Finally, we observed augmented neuronal activity in the SFO under thirst conditions after sealing CVO vessels, supported by significantly increased neuronal c-fos staining in the SFO of GOF mice.

## Results

### Wnt/β-catenin signaling is not detectable in ECs of the developing CVOs

As previously shown, BAT-gal mice report active Wnt/β-catenin in brain parenchymal vessels during embryonic and early postnatal brain vascularization (*Figure 1A*) (*Liebner et al., 2008*). So far, the developmental formation of CVOs and of the SFO in particular has not been investigated in mice. We made use of BAT-gal mice to monitor Wnt/β-catenin activity in CVO vessels at different time points of embryonic and postnatal mouse development, starting from E13.5 which was the first time-point we could identify the primordial SFO, to P21 (*Figure 1B–E*, *Figure 1—figure supplements 1* and *2*). In the SFO (*Figure 1B–E*) we could not detect an overlap of reporter gene-expression and CD31$^+$ cells at any developmental stage analyzed. Instead, adjacent, non-endothelial cells in the ependymal lining as well as cells in the stroma of the organ showed active Wnt/β-catenin signaling, evidenced by nuclear β-galactosidase staining (*Figure 1B–E*). The latter observation suggested that Wnt growth factors are indeed available in the CVO region, but the canonical pathway was not activated in ECs.

Similarly, we did also not observe reporter gene-expressing ECs in the OVLT and in the PI (*Figure 1—figure supplements 1* and *2*), providing evidence for the interpretation that CVO endothelia generally show low or no Wnt/β-catenin activity.

We further wanted to address the question, whether the lack of Wnt pathway activation in CVO vessels is evolutionary conserved and analyzed the OVLT of adult Wnt pathway reporter zebrafish (*Jeong et al., 2008*). In all fish analyzed, OVLT vessels were largely devoid of GFP reporter gene expression, suggesting that Wnt/β-catenin activation is strongly reduced or absent in this CVO of the fish (*Figure 2*).

In order to further characterize the vascular organization of the SFO, we analyzed adult wild type (WT) mice by confocal and light sheet microscopy.

### Adult SFO vessels are highly heterogenous regarding their barrier properties

As it has previously been proposed by Pócsai et al. that the SFO can be divided into a shell and a core region with different properties of astrocytes and extracellular matix (ECM), we intended to analyze the distribution of leaky and tight vessels within the SFO by staining for Plvap/Meca32 and Cldn5, respectively (*Pócsai and Kálmán, 2015*). In order to visualize the organs, relevant for water homeostasis, we initially applied fluorescent microscopy on sagittal sections, showing that indeed, vessels in the SFO, OVLT and PP were Plvap/Meca32$^+$, but also showed a considerable degree of intermingling with Cldn5$^+$ ECs (*Figure 3—figure supplement 1*).

In order to have a global view on vessel heterogeneity within the SFO, we prepared brains of WT C57Bl6 mice for whole mount staining (*Figure 3*). Light sheet microscopy analysis of whole mount preparations revealed that the majority of SFO vessels in the rostral portion, as well as of the shell were Plvap/Meca32$^-$//Cldn5$^+$, suggesting that these vessels possess BBB properties. Instead, vessels of the caudal SFO region were mainly Plvap/Meca32$^+$//Cldn5$^-$, providing evidence for their leaky phenotype (*Figure 3D*; *Video 1*). Higher magnification of the rostral part and the outer shell of the organ showed that some vessels exhibit a mosaic-like staining for Plvap/Meca32 and Cldn5 along their longitudinal extension (*Figure 3D*; *Video 1*). In order to visualize the alternating expression of leaky and tight vessel markers in more detail, we applied confocal microscopy on sagittal sections, revealing that neighboring cells may be positive either for Plvap/Meca32 or for Cldn5 (*Figure 3C*). However, some cells also showed a mixed identity, allowing the interpretation that there is a continuous transition from leaky-to-tight-to-leaky vessels in the SFO.

As we observed an alternating barrier phenotype in the SFO vasculature, we addressed the questions if dominant activation of β-catenin signaling in ECs may lead to vessel tightening of CVO vessels, particularly in the SFO.

### Dominant activation of β-catenin in ECs seals SFO vessels

To dominantly activate the Wnt/β-catenin pathway in ECs, Cdh5(PAC)-CreERT2:Ctnnb1$^{E\times 3fl/fl}$ (GOF) double-transgenic mice were induced with tamoxifen (TAM) either directly after birth for three (50

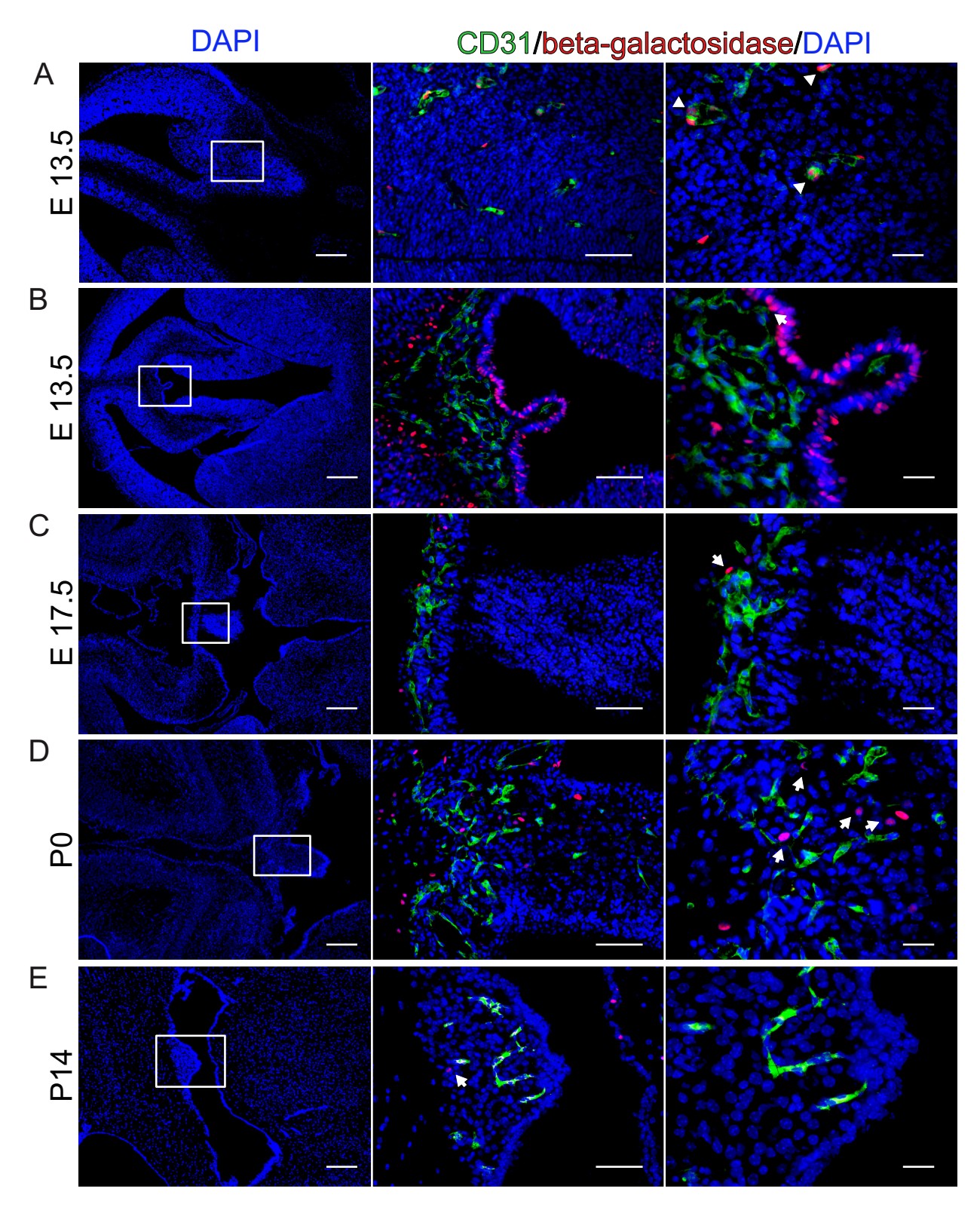

**Figure 1.** β-Catenin signaling is undetectable at different developmental stages in BAT-gal reporter mice. (**A**) Endothelial reporter gene expression, indicating β-catenin activity is detectable in cortical endothelial cells at E13.5 (arrowheads). (**B–E**) No β-catenin signaling could be detected in ECs at developmental stages E13.5, E17.5, P0 and P14 within the SFO. Arrows point to β-galactosidase positive nuclei. Scale bar: left (200 μm), middle (50 μm) and right column (20 μm).

*Figure 1 continued on next page*

*Figure 1 continued*

DOI: https://doi.org/10.7554/eLife.43818.003

The following figure supplements are available for figure 1:

**Figure supplement 1.** No endothelial β-catenin signaling in the organum vasculosum of the lamina terminalis (OVLT) during development.

DOI: https://doi.org/10.7554/eLife.43818.004

**Figure supplement 2.** No β-catenin activity in endothelial cells of the pineal gland (PI) during development.

DOI: https://doi.org/10.7554/eLife.43818.005

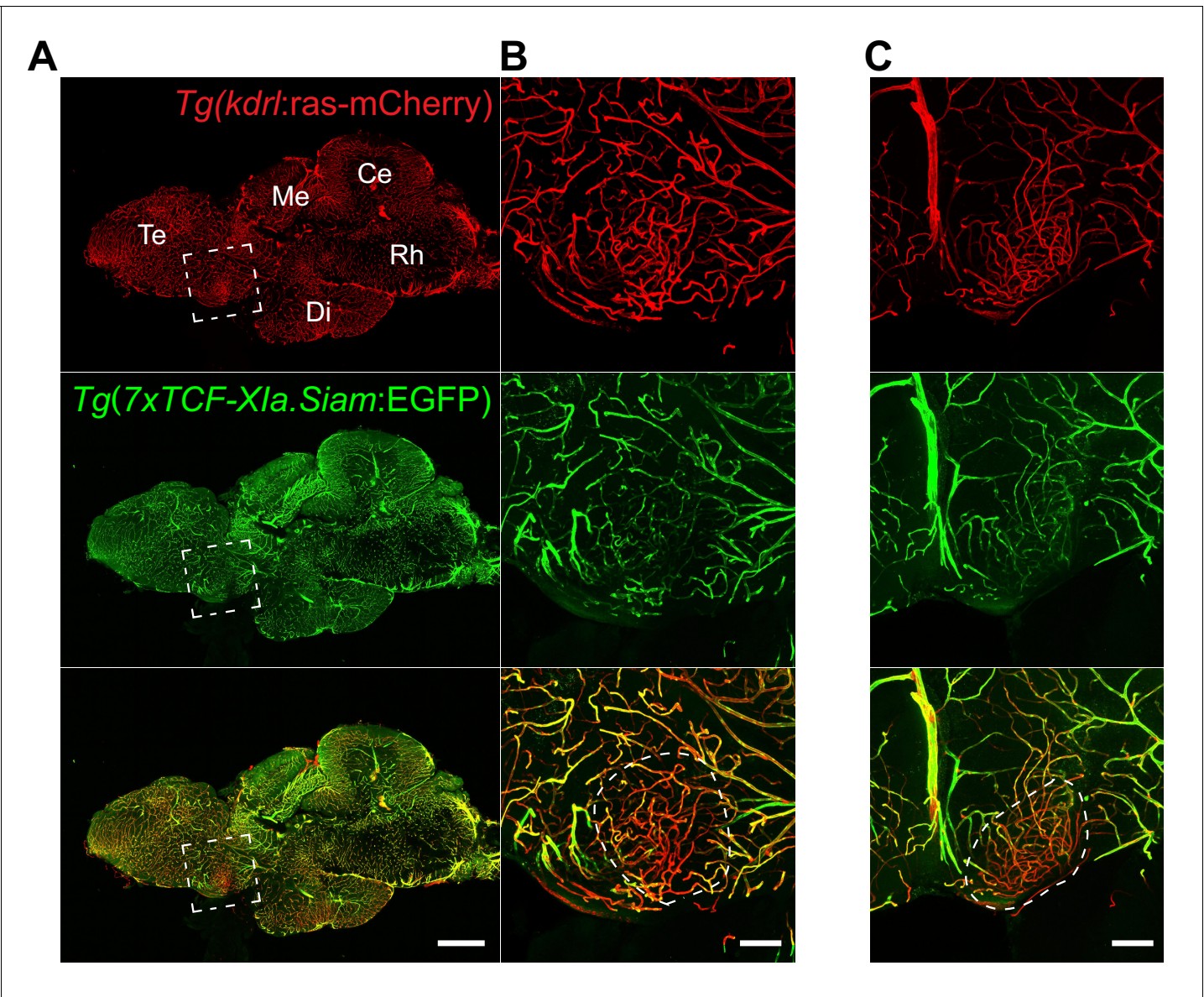

**Figure 2.** Low Wnt/β-catenin signaling in the adult zebrafish OVLT. (**A**) Midline sagittal section of an adult *Tg(kdrl:ras-mCherry):Tg(7xTCF-Xia.Siam: EGFP)* zebrafish brain. The OVLT-containing area, anatomically-defined following *Jeong et al. (2008)*, is boxed in white. (**B**) Higher magnification view of (**A**). The dense and tortuous OVLT endothelium (red) exbibits low Wnt-reporter activity (green) compared to the surrounding vessels. (**C**) Same as (**B**) in another individual. Scale bars: (**A**) 500 µm, (**B**) and (**C**) 100 µm; Te, Telencephalon; Me, Mesencephalon; Di, Diencephalon; Ce, Cerebellum; Rh, Rhombencephalon.

DOI: https://doi.org/10.7554/eLife.43818.006

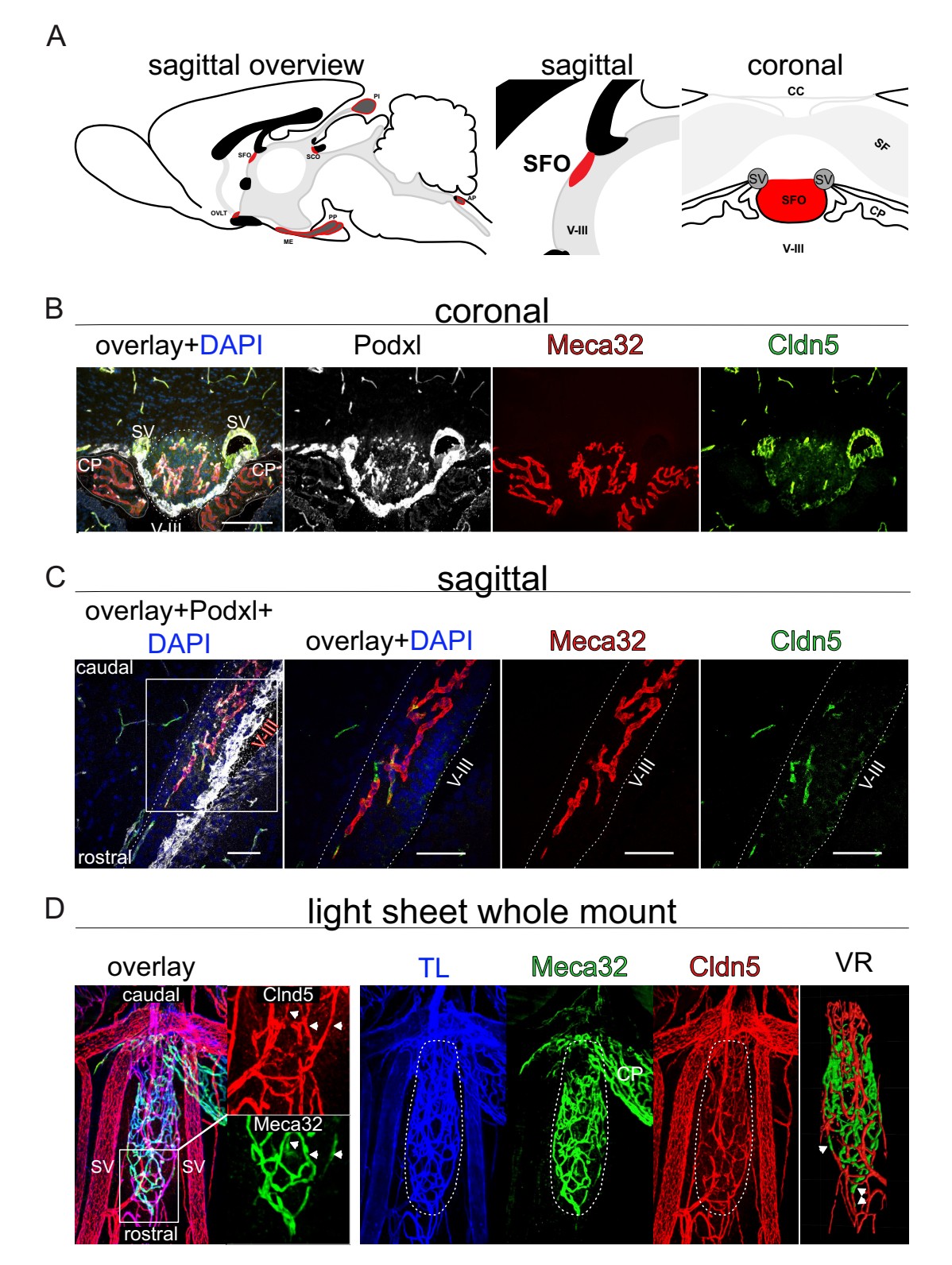

**Figure 3.** Heterogeneous barrier phenotype in vessels of the adult wild type subfornical organ (SFO). (**A**) Sagittal scheme of all circumventricular organs (CVOs) (left overview), the SFO in detail (middle sagittal, right coronal) provide an orientation. (**B**) Heterogenous barrier phenotype in coronal fluorescence images, (**C**) sagittal confocal projections of the rostral SFO tip and (**D**) light sheet projections of whole mount SFO samples with leaky MECA32[+] and tight Cldn5[+] vessels. Scale bars: (**B**) 100 µm, (**D**) first picture 50 µm and following 20 µm. SFO, Subfornical organ; OVLT, organum

*Figure 3 continued on next page*

*Figure 3 continued*
vasculosum of lamina terminalis; ME, median eminence; PP, posterior pituitary; SCO, subcommisural organ; PI, pinal organ; AP, area postrema; SV, septal veins; V-III, third ventricle; CP, choroid plexus; CC, corpus callosum; SF, septofimbral nucleus; TL, tomato lectin; VR, volume rendering.
DOI: https://doi.org/10.7554/eLife.43818.007
The following figure supplement is available for figure 3:

**Figure supplement 1.** Vessel heterogeneity in sensory circumventricular organs (CVOs) involved in water homeostasis.
DOI: https://doi.org/10.7554/eLife.43818.008

μg/day, P1-P3, *Figure 4—figure supplement 1*) or in the adult for five (500 μg/day, 8–10 week-old mice, *Figure 4*) consecutive days.

We initially determined recombination efficiency in the brain by analyzing Cdh5(PAC)-CreERT2: mTmG reporter mice (*Wang et al., 2010Muzumdar et al., 2007*), suggesting that VE-cadherin efficiently drives endothelial recombination in brain vessels (*Figure 4—figure supplement 2*).

β-Catenin GOF pups were analyzed at P6 and P14 and compared to respective controls. At both timepoints analyzed (P6 and P14), control vessels exhibited high levels of Plvap/Meca32 and low levels of Cldn5 immunolabeling (*Figure 4—figure supplement 1A,B*). Interestingly, endothelial-specific β-catenin GOF reverted this phenotype, resulting in significantly decreased Plvap/Meca32 immunoreactivity, whereas Cldn5 expression was markedly increased in these vessels without any changes in VE-cadherin immunolabeling at P6 (*Figure 4—figure supplement 1C*). In the adult, the same antagonistic regulation of Plvap/Meca32 and Cldn5 by β-catenin GOF was observed as in postnatal stages (*Figure 4B–D*; quantification 4E, F).

When analyzing different timepoints after TAM induction of adult GOF mice for the expression of Plvap/Meca32 and Cldn5, we observed that activation of β-catenin signaling significantly suppressed Plvap/Meca32 and induced Cldn5 already by day 16 (*Figure 4E,F*). Maximal Cldn5 induction was observed after 26 days, being significantly higher than at day 19 after the first TAM injection (*Figure 4E*). Analysis of pooled mRNA from 17 whole SFOs from GOF or control mice, revealed that Plvap/Meca32 was indeed down-regulated in the GOF condition on the mRNA level, whereas Cldn5 did not show an obvious regulation. This suggested that transformation of the leaky into a tight vessel phenotype in the SFO by β-catenin GOF requires around 26 days after induction of recombination.

To understand if beside Cldn5 also other tight junction components are regulated upon β-catenin activation in SFO endothelial cells, we stained for occludin (Ocln) and zonula occludens 1 (ZO-1). Analyzing the junctional localization of Ocln normalized to the vessel length in the SFO, we observed only a punctuated staining of Ocln at endothelial cell-cell junctions of controls as previously described by Morita et al. (*Morita and Miyata, 2012*). In vessels of GOF mice we noted a significant increase in line-like junctional Ocln staining compared to controls (*Figure 5A,B*; quantification *Figure 5C*). Further analysis of mRNA of pooled SFO samples from GOF or control mice, revealed that Ocln, like Cldn5, did not show an obvious regulation in the GOF condition (*Figure 5D*). As opposed to Cldn5 and Ocln, ZO-1 showed only a moderately increased localization at cell-cell junctions in SFO vessels of GOF mice (*Figure 5—figure supplement 1*). Specifically, ZO-1 was consistently present at inter-endothelial junctions of the SFO in the control condition.

As the upregulation of the junctional proteins Cldn5 and Ocln support the interpretation of an SFO vessel tightening in GOF mice, it remained to be clarified if vessel permeability is indeed affected by dominant β-catenin activation in ECs. To this end, GOF and control mice were

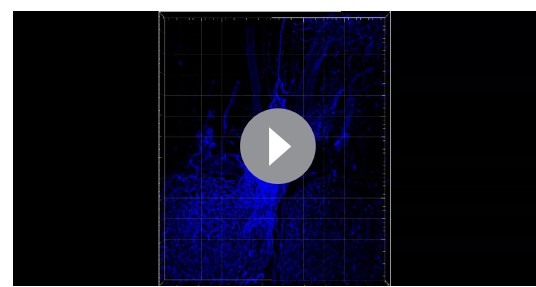

**Video 1.** Heterogeneous barrier phenotype in vessels of the adult wild type subfornical organ (SFO). Video of a cleared whole mount preparation of the SFO and neighboring tissue, stained for Cldn5 (red), Meca32/Plvap (green) and i.v.-injected tomato-lectin-Alexa649 (blue) as a general vessel marker. Volume rendering demarcates SFO vessels.
DOI: https://doi.org/10.7554/eLife.43818.009

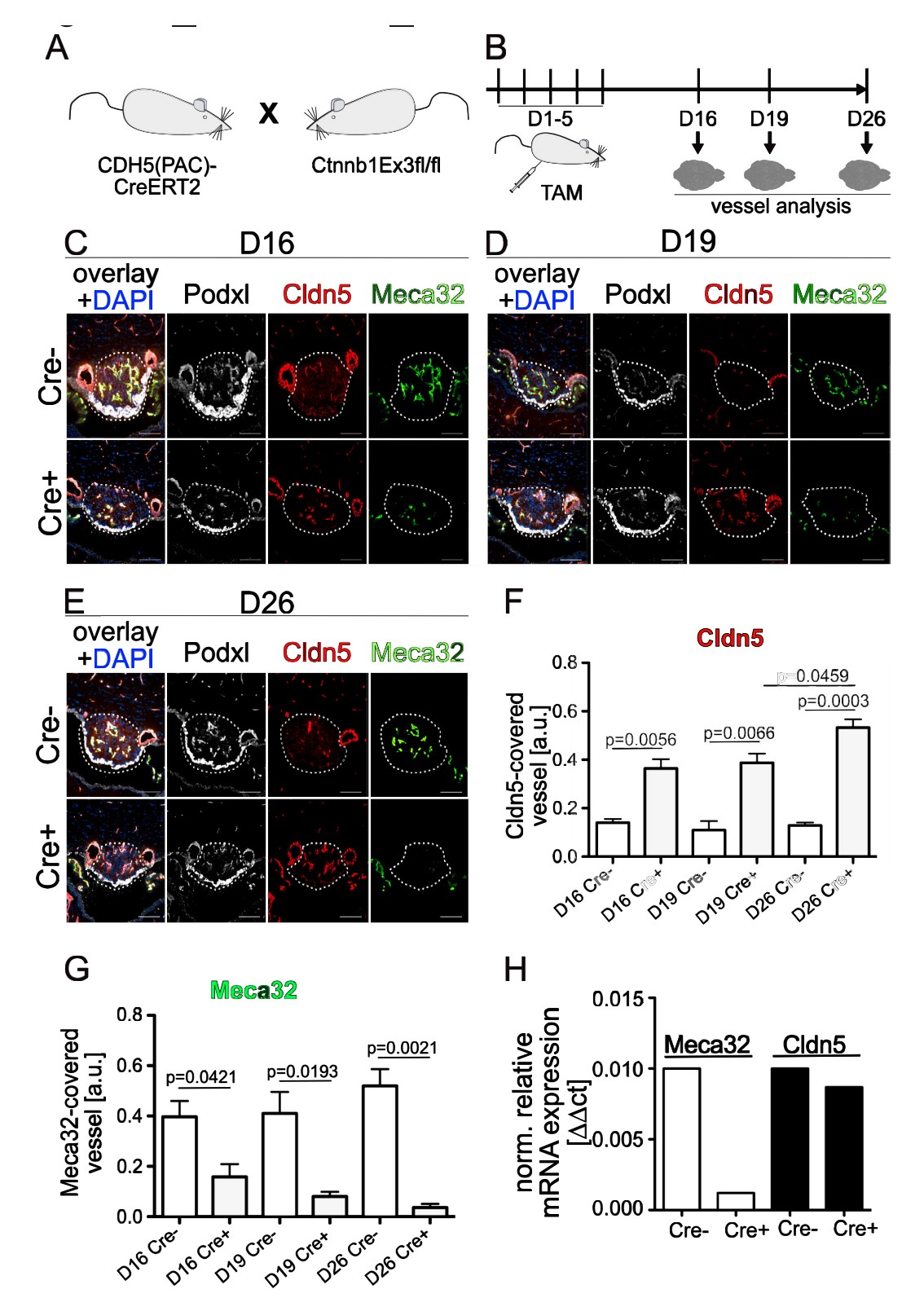

**Figure 4.** Endothelial-specific β-catenin GOF tightens the vasculature of the subfornical organ (SFO). (**A**) Mouse model and (**B**) schedule of endothelial-specific β-catenin GOF induction by tamoxifen (TAM) . Coronal view of the subfornical organ (SFO) (**C**) 16 , (**D**) 19 and (**E**) 26 days after the first TAM injection. (**F**) Quantification for Cldn5 and (**G**) Meca32-covered vessel area within the SFO (n = 3 per group). (**H**) Relative mRNA expression of SFO whole mount tissue (n = 1 of pooled samples (Cre⁻ = 18 mice, Cre⁺ = 17 mice)). Scale bars: (**C–E**) 100 μm; error bars show ±SEM.

Figure 4 continued on next page

*Figure 4 continued*

DOI: https://doi.org/10.7554/eLife.43818.010

The following source data and figure supplements are available for figure 4:

**Source data 1.** Quantification of endodthelial-specific β-catenin GOF-mediated vessel tightening in the vasculature of the subfornical organ (SFO) of Cdh5(PAC)-CreERT2:Ctnnb1Ex3fl/wt mice.

DOI: https://doi.org/10.7554/eLife.43818.016

**Source data 2.** Quantification of Meca32 and Cldn5 mRNA expression by qRT-PCR in the vasculature of the subfornical organ (SFO) of Cdh5(PAC)-CreERT2:Ctnnb1Ex3fl/wt mice.

DOI: https://doi.org/10.7554/eLife.43818.017

**Figure supplement 1.** Tightening of subfornical organ (SFO) vessels at early postnatal stages.

DOI: https://doi.org/10.7554/eLife.43818.011

**Figure supplement 1—source data 1.** Quantification of subfornical organ (SFO) vessel tightening at early postnatal stages.

DOI: https://doi.org/10.7554/eLife.43818.012

**Figure supplement 2.** Recombination of Cdh5(PAC)-CreERT2 in brain vasculature.

DOI: https://doi.org/10.7554/eLife.43818.013

**Figure supplement 3.** Pdgfb-iCreERT2:Ctnnb1fl/wt show tightening of subfornical organ (SFO) vasculature.

DOI: https://doi.org/10.7554/eLife.43818.014

**Figure supplement 3—source data 1.** Quantification of subfornical organ (SFO) vessel tightening in Pdgfb-iCreERT2:Ctnnb1Ex3fl/wt mice.

DOI: https://doi.org/10.7554/eLife.43818.015

intravenously injected with FITC-bovine serum albumin (FITC-BSA) (~68 kDa), and examined after 1.5 hrs of circulation. Analysis of FITC-BSA leakage normalized to vessel area revealed a significant reduction of extravasation in SFO vessels of the GOF versus control mice (*Figure 6*). Specifically, the leaky vessels in the controls showed pronounced FITC-BSA distribution in the circumference of vessels indicated by a prominent cloudy FITC signal in the entire SFO, whereas in the GOF condition the tracer remained confined to the vessel lumen (*Figure 6C*).

To strengthen the observation of SFO vessel tightening, we performed β-catenin GOF experiments also with the Pdgfb-iCreERT2 mouse driver line (*Claxton et al., 2008*), resulting in comparable regulation of Plvap/Meca32 and Cldn5 (*Figure 4—figure supplement 3A,B*). Activation of the Wnt/β-catenin pathway was supported by significantly increased nuclear Sox17 localization that was reported to be a downstream target of Wnt/β-catenin and to be upstream of Notch (*Corada et al., 2013*; *Zhou et al., 2015*) (*Figure 4—figure supplement 3C,D*).

Given the increase in Cldn5 expression in SFO vessels, we addressed if the endothelial tightening may also have an effect on the organization of the NVU within the core region of the SFO, in which no astrocytic endfeet are formed around vessels. Therefore, we stained GOF and control SFOs for the astrocytic endfeet markers aquaporin-4 (Aqp4), α-dystroglycan (αDag) and Kir4.1 (*Figure 7—figure supplements 1* and *2*), as well as the ECM markers laminin α 2 (Lama2) and collagen IV (ColIV) (*Figure 7—figure supplement 2*). All markers revealed the expected polarized distribution around BBB vessels in the striatum, nicely confirming staining specificity (*Figure 7—figure supplements 1A* and *2A,D*).

As previously shown for astrocytic endfeet proteins (*Pócsai and Kálmán, 2015*), leaky SFO vessels did not exhibit pronounced staining of the polarity markers αDag and Kir4.1 (*Figure 7—figure supplements 1B,C and* and *2B,C*). Moreover, none of these stainings were found to be affected by the GOF conditions, meaning that no distinct staining of vascular endfeet could be observed. Specifically, Aqp4 and αDag showed only a weak, unpolarized localization around vessels in GOF and controls, whereas the sodium channel Kir4.1 was mainly expressed by cells morphologically resembling tanycytes in the SFO (*Figure 7—figure supplements 1B,C and* and *2B,C*). The ECM components Lama2 and ColIV, revealed that in GOF and in control vessels of the SFO a vascular and an astrocytic basal lamina was present with no obvious differences in structure and distribution between conditions (*Figure 7—figure supplement 2*).

In order to further characterize the blood vessels in the SFO of β-catenin GOF mice, we employed electron microscopy to visualize their subcellular phenotype. As expected, the vessels in control SFOs showed the typical large and lacuna-like structure with an extensive ECM circumference (*Figure 7A–C*). Moreover, fenestrations were frequently observed in the control condition, a morphological feature that is consistent with a high Plvap/Meca32 expression and a permeable

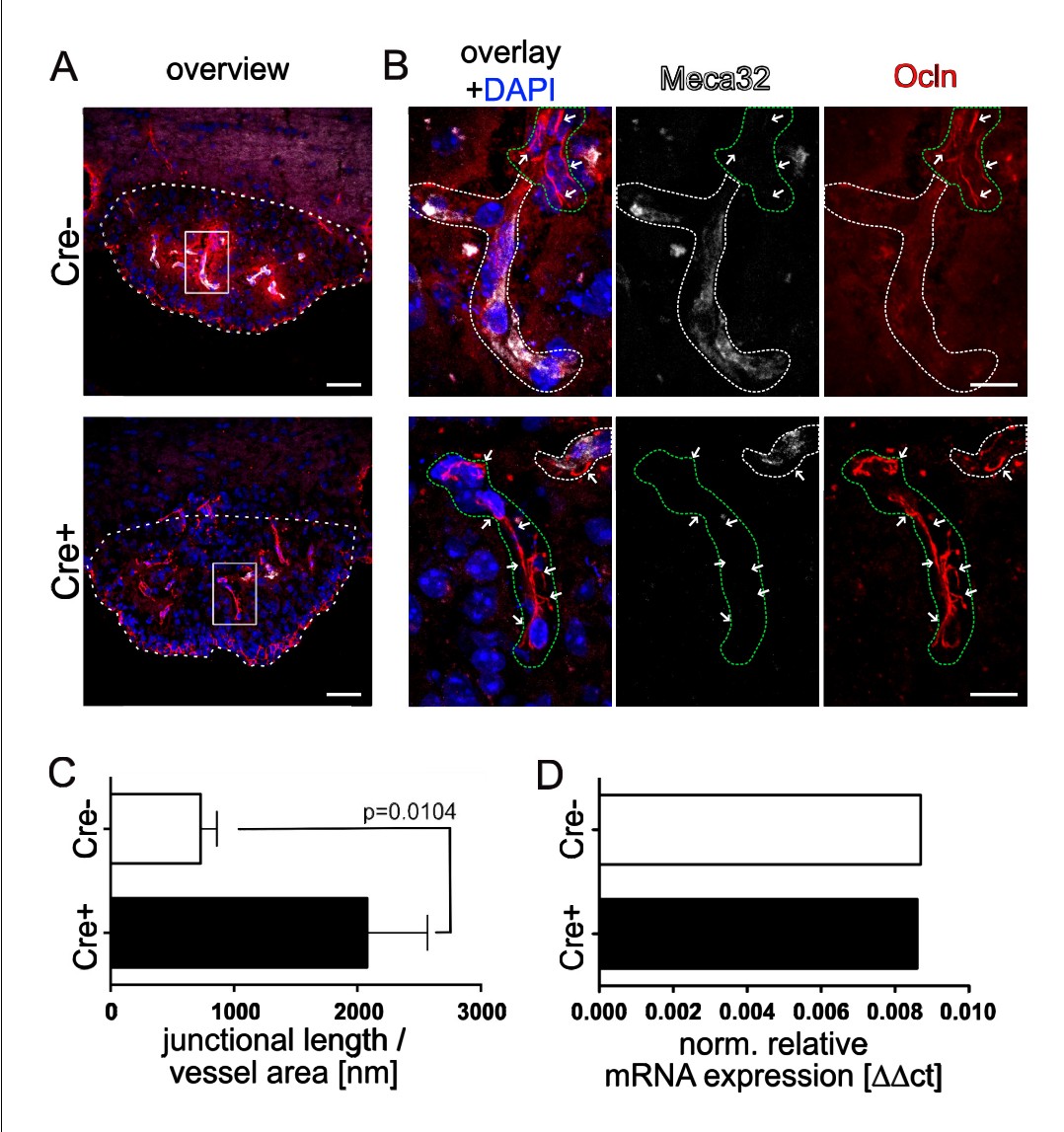

**Figure 5.** Endothelial-specific β-catenin GOF leads to increased occludin localization at cell-cell junctions in the vasculature of the subfornical organ (SFO). (**A**) Coronal view of the subfornical organ (SFO) 26 days after the first TAM injection; dashed line demarcates the SFO. (**B**) Higher magnification of an SFO vessel indicated by the rectangular inset in A, white dashed lines show Meca32[+], green dashed lines show Meca32 vessels, arrows indicate junctional Ocln staining. (**C**) Quantification for Ocln junctional length normalized to the vessel area within the SFO (n = 3 per group). (**D**) Relative mRNA expression of SFO whole mount tissue (n = 1 of pooled samples (Cre[-]=18 mice, Cre[+]=17 mice)) . Scale bars: (**A**) 50 μm, (**B**) 10 μm; error bars show ±SEM.

DOI: https://doi.org/10.7554/eLife.43818.018

The following source data and figure supplement are available for figure 5:

**Source data 1.** Quantification of occludin localization at cell-cell junctions in the vasculature of the subfornical organ (SFO) in Cdh5(PAC)-CreERT2: Ctnnb1Ex3fl/wt mice.

DOI: https://doi.org/10.7554/eLife.43818.020

**Source data 2.** Quantification of occludin mRNA expressionby qRT-PCR in the vasculature of the subfornical organ (SFO) in Cdh5(PAC)-CreERT2: Ctnnb1Ex3fl/wt mice.

DOI: https://doi.org/10.7554/eLife.43818.021

**Figure supplement 1.** Endothelial-specific β-catenin GOF did not lead to an evident increase in ZO-1 localization at cell-cell junctions in the vasculature of the subfornical organ (SFO).

DOI: https://doi.org/10.7554/eLife.43818.019

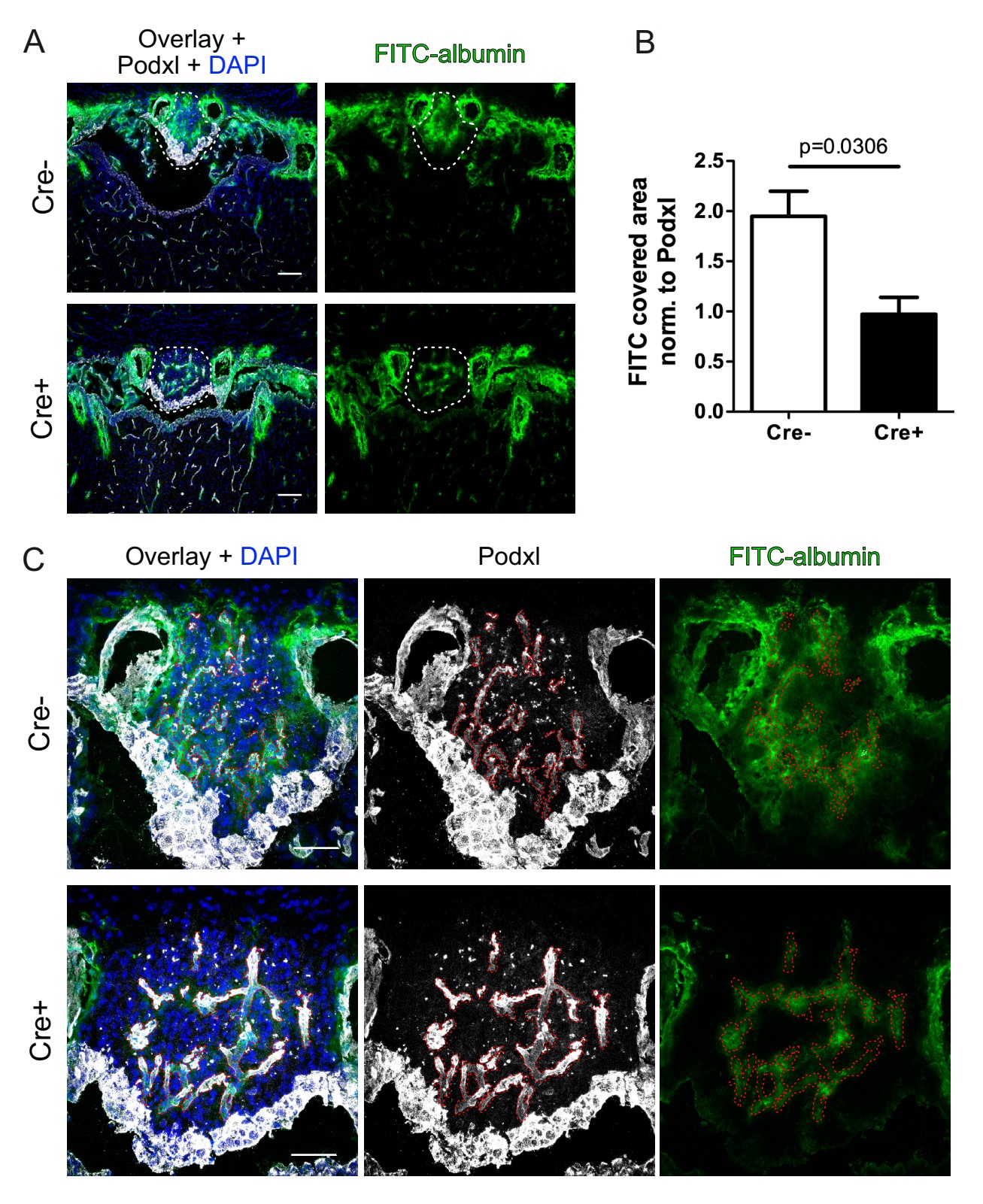

**Figure 6.** Reduction of vascular permeability by endothelial specific β-catenin gain-of-function (GOF). (**A**) Overview and (**C**) high magnification shows leakage of FITC labelled albumin within the SFO of Cre⁻ and Cre⁺ mice. Dashed lines indicate SFO (**A**) or vessel outline (**C**). (**B**) Quantification of FITC-positive SFO area normalized to the vessel area (Cre⁻ = 4 mice, Cre⁺ = 3 mice). Scale bars: (**A**) 100 μm; (**C**) 50 μm; error bars show ±SEM.
DOI: https://doi.org/10.7554/eLife.43818.022

*Figure 6 continued on next page*

*Figure 6 continued*

The following source data is available for figure 6:

**Source data 1.** Quantification of FITC-BSA extravasation in the vasculature of the subfornical organ (SFO) in Cdh5(PAC)-CreERT2:Ctnnb1Ex3fl/wt mice.
DOI: https://doi.org/10.7554/eLife.43818.023

phenotype (*Figure 7C*; quantification *Figure 7D*). Although the vessel morphology did not show major differences regarding vessel perimeter and structure, the vessels of the GOF mice appeared to have a more compact ECM deposition in their circumference (*Figure 7A–C*). Endothelial vesicles did not show obvious alterations between GOF and controls (data not shown). Instead, the junctional area of GOF vessels was considerably more elaborate compared to the controls, which exhibited typical blunt-ending connections (*Figure 7C*). Additionally, fenestrations were significantly reduced in GOF mice, being in line with the reduction of Plvap/Meca32 immunostaining (*Figure 4E*; quantification *Figure 4G*) and further suggesting that β-catenin GOF in ECs is crucial for the suppression of a leaky vessel phenotype.

Upon the observation that dominant endothelial activation of Wnt/β-catenin signaling established barrier properties in SFO vessel, we wanted to elucidate if the tightening of SFO vessels affects neuronal function in this organ.

## Endothelial β-catenin GOF results in augmented neuronal activity in the SFO of water-restricted mice

In order to understand if dominant activation of β-catenin signaling in ECs of the SFO may influence neuronal activity in the context of water homeostasis and drinking behavior, we induced thirst in adult mice and analyzed neuronal activity. To this end, WT mice were either kept for 72 hrs under water restriction (*Figure 8A*) or were intraperitoneally injected with a hyperosmolar NaCl (3 M) solution 50 min prior to sample collection (*Figure 8B*) and subsequent assessment of neuronal activation by c-fos staining in the SFO (*Figure 8C,D*). The Nissl staining of the so-called Nissl flounders nicely documents the neuronal identity of the c-fos$^+$ cells (*Figure 8*). As opposed to the general nuclear staining by the fluorescent Nissl stain, the flounders are specific for neurons only.

Both thirst-inducing paradigms lead to a significant increase in c-fos$^+$ neurons in the SFO of WT mice (*Figure 8*). We could also show a dose-dependent c-fos activation in thirst induction by hyperosmolar NaCl, comparing 2 M and 3 M solutions (*Figure 8—figure supplement 1*). Given that water restriction is a more physiological setting which reflects the restricted availability of resources in nature, we made use of this paradigm to investigate the influence of β-catenin GOF on neuronal activity in the SFO.

Under control conditions, in which mice received water *ad libitum*, we could not detect any genotype-specific differences in c-fos$^+$ nuclei in the SFO between control and GOF animals (data not shown). β-Catenin GOF and control mice were subjected to water restriction 26 days after induction by TAM (*Figure 9A*). In case of water restriction for 72 hrs (*Figure 9A*) a slight, but stable weight loss was induced in GOF and control mice in the same manner (*Figure 9B*). Analysis of c-fos activation revealed a significantly higher neuronal activity in the SFO of GOF mice (*Figure 9D*). This suggests that tightening SFO blood vessels may have physiological consequences for the water homeostasis in mice.

## Discussion

The present study deals with the regulation of the leaky vascular phenotype in the CVOs and in the SFO in particular. Specifically, we addressed the questions, a) if the Wnt/β-catenin pathway is operational in ECs of CVOs during murine development and in the adult mouse and zebrafish, b) if endothelial-specific, dominant activation of β-catenin transcription could convert the leaky vascular phenotype in CVOs and c) if the latter may have an effect on CVO function.

The principle findings of this study are: 1) Wnt/β-catenin signaling is undetectable in CVO vessels during BAT-gal reporter mouse development; 2) similarly, β-catenin-mediated transcription is strongly reduced in the adult zebrafish OVLT; 3) SFO vessels are heterogenous regarding the expression of Plvap/Meca32 and Cldn5; 4) upon genetic β-catenin GOF in ECs, leaky SFO vessels

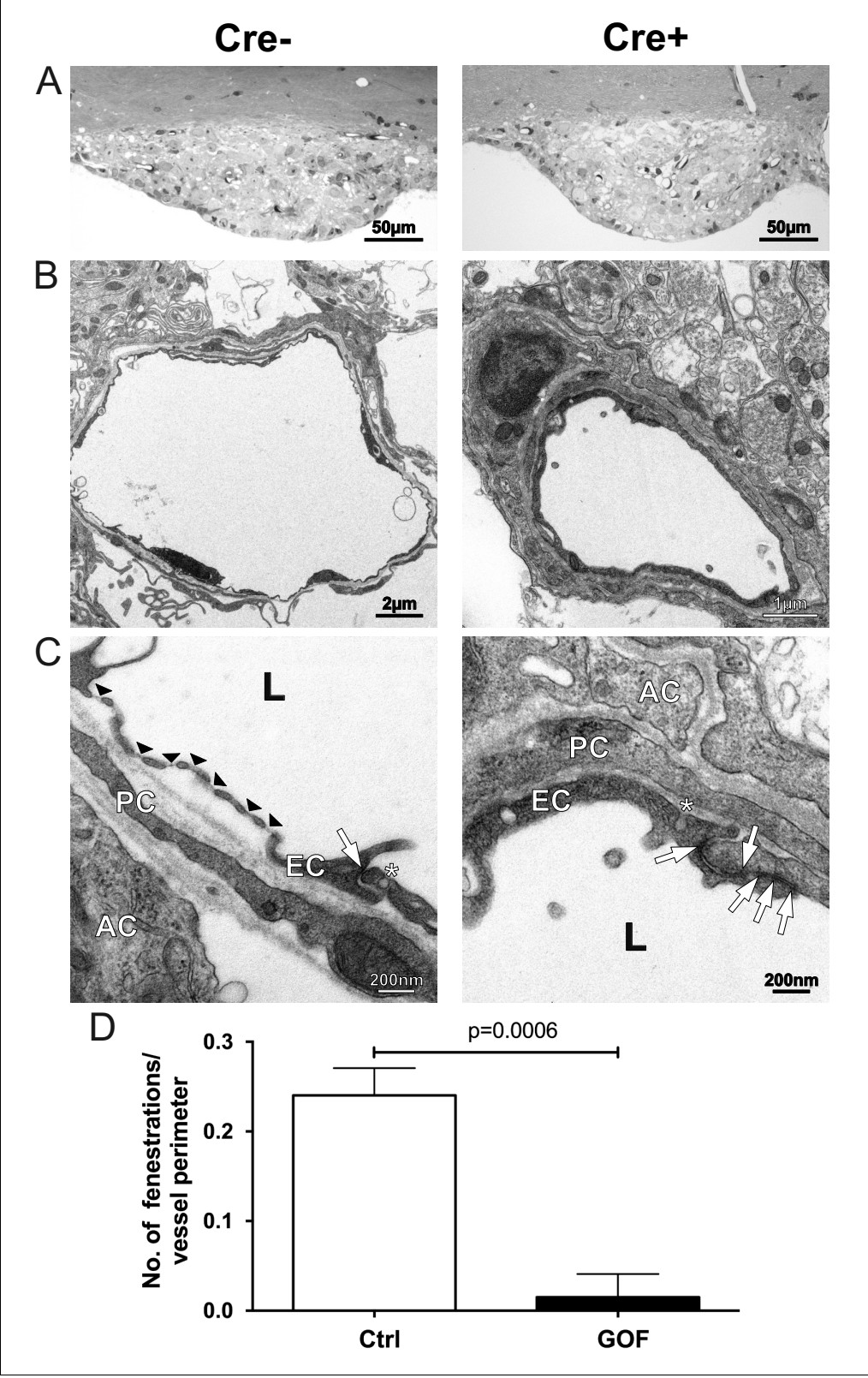

**Figure 7.** Tightening of vessels in the subfornical organ (SFO) on cellular level. (**A**) Semithin sections of SFO of endothelial-specific β-catenin GOF (Cre+) and controls (Cre-). Electron microscopic picture of Cre- (**B**), (**C**), left column) and Cre+ (**B**), (**C**), right column). Black arrow heads indicate fenestrations, with arrows endothelial

*Figure 7 continued on next page*

*Figure 7 continued*

junctions, asterisks show vesicles. AC, astrocyte; EC, endothelial cell; L, lumen; PC, pericyte. (**D**) Number of fenestrations are quantified in three vessel sections per animal (n = 4). Error bars show ±SEM.

DOI: https://doi.org/10.7554/eLife.43818.024

The following source data and figure supplements are available for figure 7:

**Source data 1.** Quantification of endothelial fenestrations in the vasculature of the subfornical organ (SFO) in Cdh5(PAC)-CreERT2:Ctnnb1Ex3fl/wt mice.

DOI: https://doi.org/10.7554/eLife.43818.027

**Figure supplement 1.** Endothelial β-catenin GOF does not affect astrocytic endfoot polarization of α-dystroglycan (α-Dag) and Kir4.1 within the subfornical organ (SFO).

DOI: https://doi.org/10.7554/eLife.43818.025

**Figure supplement 2.** Endothelial β-catenin GOF does not affect the ECM of astrocytic endfeet and ECs within the subfornical organ (SFO).

DOI: https://doi.org/10.7554/eLife.43818.026

are partially converted into tight vessels; 5) functional conversion of SFO vessel towards a BBB-like identity affects neuronal activity in the SFO.

Wnt/β-catenin is crucial for brain vascularization and BBB development, by regulating endothelial sprouting as well as by promoting a BBB expression profile in ECs, respectively (*Vanhollebeke et al., 2015*; *Liebner et al., 2008*; *Daneman et al., 2009*; *Stenman et al., 2008*; *Zhou et al., 2014*). CVOs are well known, but poorly investigated, structures in the midline of vertebrate brains, conferring neurosensory and/or neurosecretory function. Because of this physiological function, CVO blood vessels were described for a long time to lack BBB characteristics, a feature that is considered to be important for allowing neurons to 'sense' salts, hormones, lipids and toxic compounds in the blood (*Sisó et al., 2010*; *Kiecker, 2018*). Indeed, it has been shown that neurons send axons into the extended perivascular space, which is in line with their sensory function (*Morita and Miyata, 2012*). The peculiar, leaky specialization of the CVO vascular system is well documented, showing tortuous and fenestrated vessels with poorly developed inter-endothelial junctions (*McKinley et al., 2003*). However, how this specialization is induced on a molecular level during development and how it is maintained is currently not well understood. Vascular endothelial growth factor (VEGF) is the best described inducing factor for endothelial fenestrations and is reported to be expressed in sensory CVOs as well as in other tissues that physiologically require endothelial fenestrations, such as the choroid and the ciliary body of the eye (*Furube et al., 2014*; *Ford et al., 2012*; *Kinnunen and Ylä-Herttuala, 2012*).

As the Wnt/β-catenin pathway is considered a master switch for barriergenesis, we hypothesized that β-catenin transcription is not operational during CVO vascularization. The data provided here support this interpretation, as in BAT-gal reporter mice, from the initial identification of the SFO primordium at E13.5, none of the investigated developmental stages revealed a single β-galactosidase-positive vessel within the CVOs (*Figure 1*; *Figure 1—figure supplements 1* and *2*). Although this finding may formally not exclude low level activation of the pathway in ECs, the observation that neighboring, non-endothelial cells in the CVOs, do show Wnt pathway activation, supports the interpretation of low or absent Wnt/β-catenin signaling in CVO ECs. Specifically, we observed that the ependymal cells covering the SFO as well as stromal cells in the core of the organ show Wnt/β-catenin pathway activation (*Figure 1*). Quantitative RT-PCR revealed also expression of Wnt3a, Wnt7 as well as Fzd4 in the SFO, suggesting that at least the BBB-inducing machinery is expressed (data not shown). As also in the adult, β-catenin-mediated transcription in ECs is required to maintain BBB function, the absence of reporter activity in mouse (data not shown) or zebrafish models presented in this study, further underlines that Wnt is evidently not operational in CVO vessels. This may support the hypothesis that Wnt/β-catenin signaling is actively suppressed in the SFO and likely also in other CVOs that have fenestrated vessels. So far, no conclusive data are available demonstrating Wnt pathway inhibitors in the CVOs, however, it has been shown that expression of the soluble frizzled receptor protein 1 (Sfrp1) is about 30 times higher in the rat choroid plexus (CP), that also lacks BBB vessels, compared to the striatum and parietal cortex (*Bowyer et al., 2013*). Nevertheless, a detailed analysis of the CVOs regarding cell type-specific expression profiles has not been published yet.

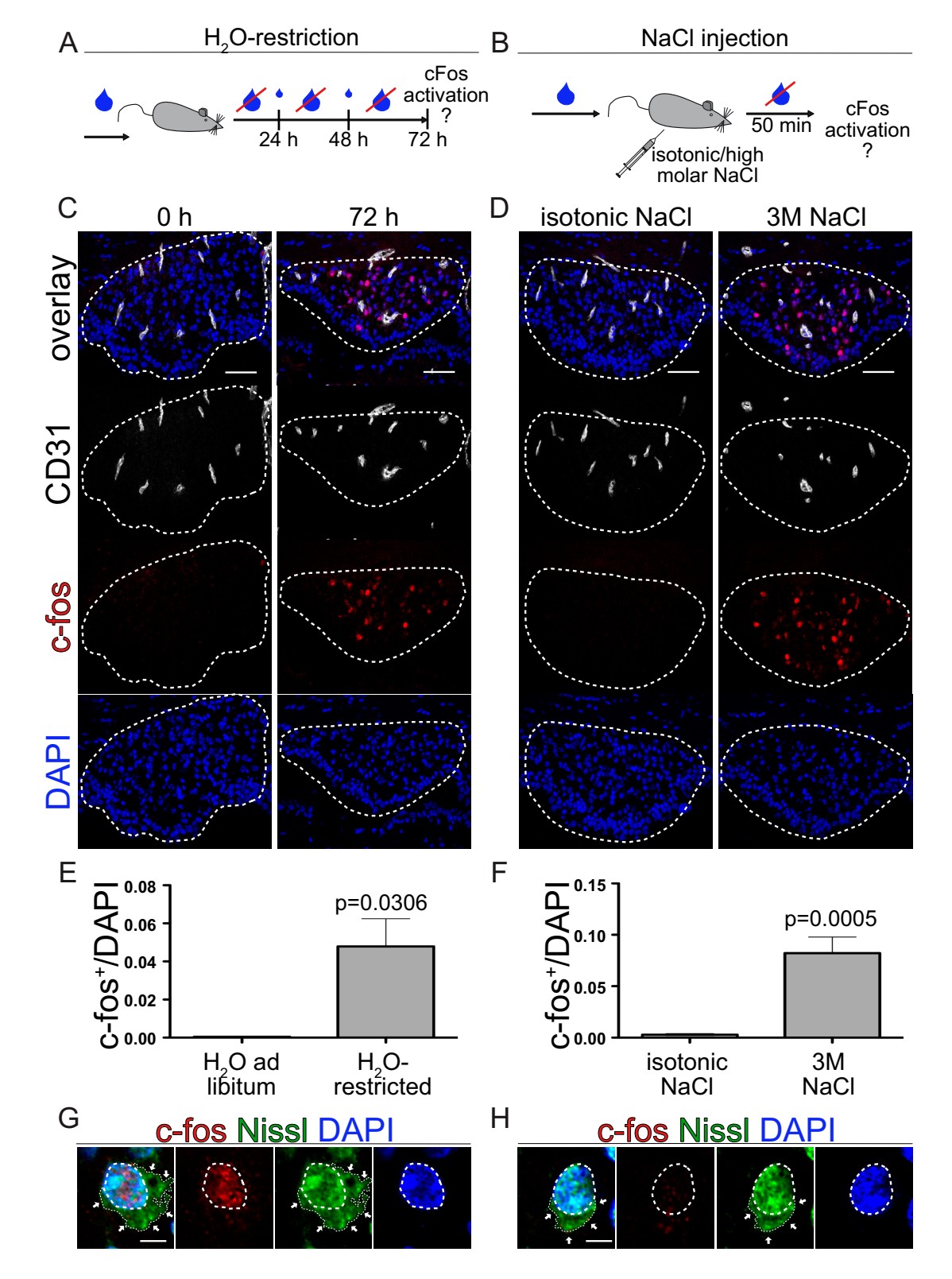

**Figure 8.** Neuronal activation via thirst induction in wild type animals. (A) Schedule of water restriction paradigm. Small blue droplets represent a restricted amount of water in a 24 hrs cycle according to the bodyweight (BW). (C) c-fos activation in the SFO of mice with water *ad libitum* and animals restricted for 72 hrs. (E) quantification of c-fos positive/DAPI nuclei in the SFO (n = 3). (B) Experimental setting of hyperosmolar sodium chloride injection. Animals get either isotonic (0.15 M) or 3 M sodium chloride intraperitoneally injection (150 μl/20 g mouse). c-fos analysis 50 min after NaCl

*Figure 8 continued on next page*

*Figure 8 continued*

injection (D) and quantification (F) (n = 6). Dashed lines indicate Nissl flounders confirming neuronal idendity of c-fos⁺ (G) and c-fos⁻ (H) cells. Scale bars: (C), (D) 50 µm, (G), (H) 2 µm; error bars show ±SEM.

DOI: https://doi.org/10.7554/eLife.43818.028

The following source data and figure supplements are available for figure 8:

**Source data 1.** Quantification of dose dependent neuronal activity in the subfornical organ (SFO) upon hyper-osmolar sodium chloride injection.
DOI: https://doi.org/10.7554/eLife.43818.031

**Figure supplement 1.** Dose dependent neuronal activity upon sodium chloride injection.
DOI: https://doi.org/10.7554/eLife.43818.029

**Figure supplement 1—source data 1.** Quantification of neuronal activation in the subfornical organ (SFO) via thirst induction in wild type mice.
DOI: https://doi.org/10.7554/eLife.43818.030

Still, it has to be noted that vessels in the SFO, OVLT and PP are heterogenous regarding the expression of Plvap/Meca32 and Cldn5 (*Figure 3*, *Figure 3—figure supplement 1*). Similarly, differentially tight vessels were also shown in other CVOs (*Morita and Miyata, 2012*). This raises the question if in the CVOs, unlike in the brain parenchyma a '...gradual phenotypic change (zonation) along the arteriovenous axis...' is realized (*Vanlandewijck et al., 2018*), or if alternating endothelial differentiation might be established by factors yet to be discovered.

The present findings may suggest that, at least to some degree, vascular phenotypes in the SFO are locally regulated, which would be in line with their role providing local access for neurons to the blood milieu. If vessel differentiation might also be dynamically regulated to control water homeostasis in a circadian rhythm (*Gizowski et al., 2016*), is currently unknown and subject to ongoing investigation. In this regard it is interesting to note however, that Cldn5 and Ocln mRNA were not significantly upregulated when analyzed in whole mount dissected SFOs from GOF mice (*Figures 4H* and *5D*). This might be due to several reasons, such as signal masking by other vessels in the whole mount preparations. Alternatively, this finding might support the interpretation that Cldn5 and Ocln are not transcriptionally regulated by β-catenin, but rather regulated on a post-transcriptional level. Interestingly, there is still some controversy about Cldn5 regulation by Wnt/β-catenin, as it has been shown by Taddei et al. that β-catenin cooperates with FOXO1 to suppress Cldn5 at the promotor level under pro-angiogenic conditions (*Taddei et al., 2008*). On the other hand, it has been shown that Sox18, a member of the SOX family of high-mobility group box transcription factors, is instrumental in activating Cldn5 transcription, contributing to endothelial barrier formation (*Fontijn et al., 2008*). Given the high redundancy of SoxF genes (Sox7, 17, 18) (*Zhou et al., 2015*), it might be feasible that Sox17, that we report here to be upregulated in SFO vessels of β-catenin GOF mice (*Figure 5—figure supplement 1*), mediates Cldn5 regulation.

Although the regulation of Cldn5 on the promotor level and the role of β-catenin herein requires additional investigation, the tightening of SFO vessels by Cldn5 protein upregulation in β-catenin GOF mice is consistent with previous reports in other regions of the brain (*Zhou et al., 2014*). Interestingly, in the SFO of GOF mice, we also observed a significantly augmented junctional localization of Ocln, which is in line with the endothelial tightening, but, like for Cldn5, at which molecular level the Ocln regulation occurs remains to be clarified. Moreover, the adherens and tight junction-associated protein ZO-1 qualitatively showed a slight increase in junctional continuity in the GOF condition, which fits with the overall formation of more elaborate junctional complexes between ECs. The fact that ZO-1 exhibits also junctional staining in the controls (*Figure 5—figure supplement 1*), is consistent with its role in VE-cadherin-based adherens junctions, which are also formed by SFO vessels (*Figure 4—figure supplement 1*) (*Tornavaca et al., 2015*).

Beside the mere upregulation of Cldn5 and Ocln, endothelial β-catenin GOF resulted in the abolishment of fenestrations and strengthened inter-endothelial junctions. These findings are well in line with a reduction in VEGF signaling in glioma ECs upon Wnt/β-catenin activation via the downregulation of VEGF receptor 2 (VEGFR2, flk-1) and upregulation of VEGFR1 (*Reis et al., 2012*). This suggests that also upon β-catenin GOF in CVO vessels the responsiveness of ECs for VEGF could be reduced, leading to regression of fenestrations. Interestingly, the structural components of the NVU such as astrocytic endfeet and ECM, additional crucial BBB features, were not observed to be changed by the GOF condition (*Figure 7—figure supplements 1* and *2*). Also, the vessel coverage by pericytes showed no major changes in the SFO comparing GOF and controls (data not shown). If

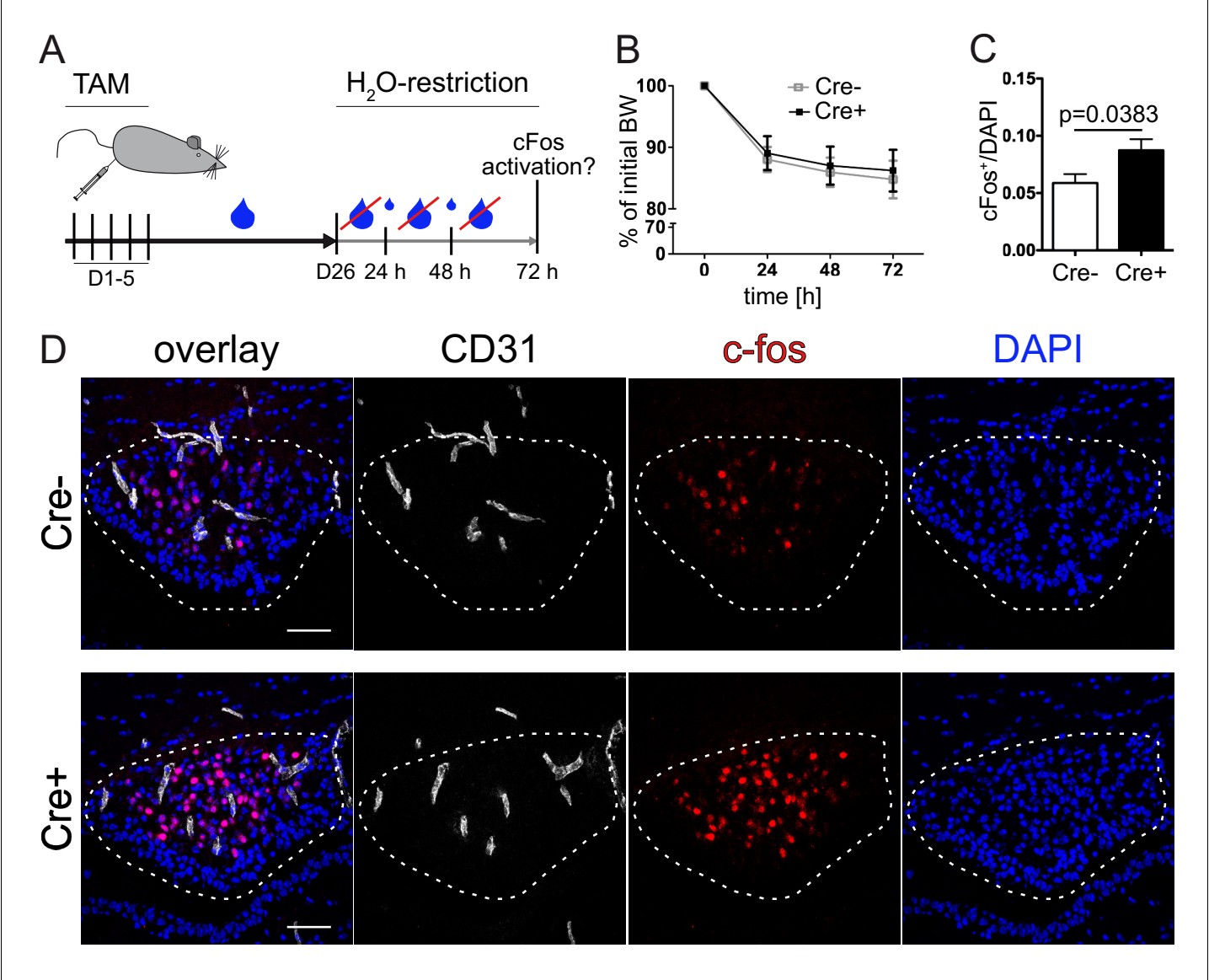

**Figure 9.** Vascular tightening effects increased neuronal activity in the subfornical organ (SFO) under thirst conditions. (**A**) Experimental setup of water restriction in β-catenin GOF and control mice after tamoxifen (TAM) injection. (**B**) Monitoring of BW for GOF and control mice under water restriction. (**D**) c-fos activation (dashed lines indicate the SFO) and (**C**) quantification of c-fos positive/DAPI nuclei in the SFO (n(Cre⁻) = 9, n(Cre⁺) = 8). Scale bars show 50 μm; error bars show ±SEM.

DOI: https://doi.org/10.7554/eLife.43818.032

The following source data is available for figure 9:

**Source data 1.** Quantification of neuronal activity in the subfornical organ (SFO) under thirst conditions in Cdh5(PAC)-CreERT2:Ctnnb1Ex3fl/wt mice.
DOI: https://doi.org/10.7554/eLife.43818.033

the perivascular fibroblasts, recently described by Vanlandewijck et al. (*Vanlandewijck et al., 2018*), are present at SFO vessels and if yes, whether they are affected by β-catenin GOF in ECs has to be determined in future investigations. Hence, to form the NVU structure might require additional cues and/or prolonged time to form, although the latter explanation might not be as likely as the first, given that even after sixty days after TAM injection the control-like phenotype persisted (data not shown). These findings support the conclusion that tightening the ECs in the SFO via β-catenin GOF does not lead to pronounced structural alterations at the NVU. As the leaky vessels of the SFO core are surrounded by a prominent perivascular space, which is considered to be important for the

communication of neuronal axons with the blood milieu, it might be therapeutically beneficial that the NVU is not affected by the dominant activation of endothelial β-catenin.

One of the main questions investigated in the present work is how the vasculature in the sensory CVOs like the SFO functionally cooperates with the neurons and other stromal cells to achieve proper physiological regulation of fundamental body parameters like water homeostasis. So far, the vasculature has drawn little attention in this respect, even though considerable progress has recently been made to unravel the regulation of drinking behavior by the SFO, OVLT and the PP (*Gizowski et al., 2016*; *Matsuda et al., 2017*; *Oka et al., 2015*; *Zimmerman et al., 2016*; *Augustine et al., 2018*). Specifically, it was shown for the SFO that two distinct populations of neurons expressing ETV-1 and Vgat mediate thirst-ON and thirst-OFF signals, respectively (*Oka et al., 2015*). Here we provide evidence for an essential role of endothelial barrier function in neuronal activation in water restricted mice, as neuronal c-fos reactivity was increased in water-deprived GOF animals (*Figure 9*). How this finding relates to the drinking behavior and to the activity of excitatory and inhibitory neuronal signals to and from the median preoptic nucleus (MnPO), which was shown to host the behavioral output neurons (*Augustine et al., 2018*), is beyond the scope of this study and is subject to future work. Moreover, it remains to be clarified if the increased c-fos signal in the SFO of GOF mice is directly caused by the tightened vessel phenotype, or indirectly affected by an altered angiocrine profile of the tightened endothelium, potentially leading to altered drinking behavior. Preliminary analysis of primary mouse brain microvascular ECs (MBMECs) treated with Wnt3a revealed no regulation of VEGF that was previously described to be neuroactive (data not shown) (*Mackenzie and Ruhrberg, 2012*).

Although own preliminary experiments aiming to pharmacologically tighten CVO vessels with a systemically administered Wnt/β-catenin activator did not result in SFO vessel tightening (data not shown), this might be a potential way to therapeutically modulate water intake. Interestingly, patients that chronically receive LiCl, an FDA-approved drug for bipolar disorders and a potent Wnt/β-catenin activator, frequently develop polyuria that is linked to altered anti-diuretic hormone (ADH; vasopressin) function, which is released by the PP. Moreover, many patients develop polydipsia and urinate more frequently (*Malhi, 2015*). Hence in-depth investigation of the pharmacologic modulation of SFO vessel permeability is required. Although the detailed mechanisms underlying the neuro-vascular coupling in the CVOs have to be investigated in more detail, in light of the present work however, the CVO vasculature likely participates actively in controlling water homeostasis.

# Materials and methods

### Key resources table

| Reagent type (species) or resource | Designation | Source or reference | Identifiers | Additional information |
|---|---|---|---|---|
| Strain, strain background (*Mus musculus*) | Wild-type mice | ENVIGO, The Netherlands | C57BL/6J | |
| Strain, strain background (*Mus musculus*) | Cdh5-cre mice | Ralf H. Adams, Max-Planck-Institute for Molecular Biomedicine, Münster, Germany | Cdh5(PAC)-CreERT2 | |
| Strain, strain background (*Mus musculus*) | Pdgfb-cre mice | Marcus Fruttiger (University College London, London, UK | PDGFB-iCreERT2 | |
| Strain, strain background (*Mus musculus*) | β-Catenin exon3 -floxed mice | M. Mark Taketo, Kyoto University, Japan | Ctnnb1$^{Ex3fl/fl}$ | |
| Strain, strain background (*Mus musculus*) | Wnt/β-catenin reporter mice | Stefano Piccolo (University of Padua, Padova,Italy) | B6.Cg-Tg(BAT-lacZ)3Picc/J | |

*Continued on next page*

*Continued*

| Reagent type (species) or resource | Designation | Source or reference | Identifiers | Additional information |
|---|---|---|---|---|
| Strain, strain background (*Mus musculus*) | Cre-reporter mice | Liqun Luo, Stanford University | STOCK *Gt(ROSA)26Sor*<sup>tm4(ACTB-tdTomato,-EGFP)Luo</sup>/J | |
| Strain, strain background (*Danio rerio*) | Vessel reporter fish | D.Y.R. Stainier, Max-Planck-Institute for Heart and Lung Research, Bad Nauheim, Germany | *Tg(kdrl:Hsa.HRAS-mCherry)*<sup>s896</sup> | |
| Strain, strain background (*Danio rerio*) | Wnt/β-catenin reporter fish | Francesco Argenton (University of Padua, Padova, Italy) | *Tg(7xTCF-Xla.Siam:GFP)*<sup>ia4</sup> | |
| Antibody | Anti-aquaporin 4 (Aqp4; rabbit, polyclonal) | EMD Millipore | AB 2218, RRID: AB_11210366 | 1:200 PFA fixation |
| Antibody | Anti-β-galactosidase (βGal; rabbit, polyclonal) | MP Biomedicals | #55978 | 1:1000 PFA |
| Antibody | Anti-PECAM/CD31 (rat, monoclonal) | BD Pharmingen | #553370, RRID: AB_394816 | 1:100 PFA |
| Antibody | Anti-Cdh5/ VE-Cadherin (goat, polyclonal) | Santa-Cruz Biotechnology | sc-6458, RRID: AB_2077955 | 1:50 PFA/ MetOH |
| Antibody | c-fos (H-125) (rabbit, polyclonal) | Santa-Cruz Biotechnology | sc-7202, RRID: AB_2106765 | 1:1000 PFA |
| Antibody | Anti-claudin-5/Cldn5 (rabbit, polyclonal) | Thermo Fisher Scientific | #341600 | 1:200 PFA/ MetOH |
| Antibody | Anti-Collagen IV (rabbit, polyclonal) | BioRad | #2150–1470, RRID: AB_2082660 | 1:300 PFA |
| Antibody | Anti-α-dystroglycan/ α-Dag (mouse, monoclonal) | Novus-Biologicals | NBP1-49634, RRID: AB_11015510 | 1:50 PFA* |
| Antibody | Anti-Kir4.1 (rabbit, polyclonal) | Alomone labs | APC-035, RRID: AB_2040120 | 1:200 PFA* |
| Antibody | Anti-Laminin α 2/Lama2 (rat, monoclonal) | Abcam | ab11576, RRID: AB_298180 | 1:200 MetOH |
| Antibody | Anti-occludin (mouse, monoclonal) | Thermo Fisher (Invitrogen) | #33–1500, RRID: AB_2533101 | 1:100 PFA* |
| Antibody | Anti-Plvap/ Meca32 (rat, monoclonal) | BD Pharmingen | #553849, RRID: AB_395086 | 1:100 PFA/ MetOH |
| Antibody | Anti-podocalyxin/Podxl (goat, polyclonal) | R and D Systems | AF1556, RRID: AB_354858 | 1:100 PFA/ MetOH |
| Antibody | Anti-Sox17 (goat, polyclonal) | R and D Systems | AF1924, RRID: AB_355060 | 1:100 PFA |
| Antibody | Anti-ZO-1 (rabbit, polyclonal) | Thermo Fisher (Invitrogen) | #40–2300, RRID: AB_2533457 | 1:100 MetOH |

*Continued on next page*

*Continued*

| Reagent type (species) or resource | Designation | Source or reference | Identifiers | Additional information |
|---|---|---|---|---|
| Antibody | Anti-goat IgG DyLight 550-conjugated (donkey, polyclonal) | Thermo Fisher Scientific | SA5-10087, RRID: AB_2556667 | 1:500 PFA/MetOH |
| Antibody | Anti-goat IgG DyLight 650-conjugated (donkey, polyclonal) | Thermo Fisher Scientific | SA5-10089, RRID: AB_2556669 | 1:500 PFA/MetOH |
| Antibody | Anti-rabbit IgG DyLight 488-conjugated (donkey, polyclonal) | Thermo Fisher Scientific | SA5-10038, RRID: AB_2556618 | 1:500 PFA/MetOH |
| Antibody | Anti-rabbit IgG DyLight 550-conjugated (donkey, polyclonal) | Thermo Fisher Scientific | SA5-10039, RRID: AB_2556619 | 1:500 PFA/MetOH |
| Antibody | Anti-rat IgG DyLight 550-conjugated (donkey, polyclonal) | Thermo Fisher Scientific | SA5-10027, RRID: AB_2556607 | 1:500 PFA/MetOH |
| Other | DAPI | Molecular Biological Technology (Mo Bi Tec) | D-1306 | 300 µM (1:800) |
| Other | NeuroTrace™ Green Fluorescent Nissl Stain | Thermo Fisher Scientific | N21480 | 1:300 PFA |
| Other | Tissue-Tek O.C.T. | Sakura Finetek Europe | 4583 | |
| Other | qPCR SYBR Green Fluorescein Mix | Thermo Fisher Scientific | AB-1219 | |
| Chemical compound | Tricaine methanesulfonate (MS-222) | Sigma-Aldrich | E10521 | |
| Chemical compound | TAM | Sigma-Aldrich | T5648 | |
| Chemical compound | FITC-albumin | Sigma-Aldrich | #A9771 | |
| Chemical compound | tomato lectin Alexa 649 | Vector laboratories | (#DL-1178 | |
| Chemical compound | ethylcinnamate (ECi),) | Sigma-Aldrich | (#112372 | |
| Chemical compound | AR6 Buffer | Perkin Elmer | (#AR600250ML | |
| Commercial kit | RNeasy plus Micro kit | Qiagen | #74034 | |
| Commercial kit | RevertAidTM H minus first strand cDNA synthesis kit | Thermo Fisher Scientific | #K1632 | |
| Commercial kit | RNeasy Mini kit | Quiagen | #74104 | |

## Animal models

Mice were housed under standard conditions with 12 hrs light dark cycle and water and mouse chow available *ad libitum* if not declared otherwise. All experimental protocols, handling and use of mice were approved by the Regional Council Darmstadt, Germany (V54-19c20/15-FK/1052 and V54-19c20/15-FK/1108). Wildtype (WT) C57BL6/J as well as transgenic animals were used. The following mouse strains were included Cdh5(PAC)-CreERT2 (*Wang et al., 2010*), PDGFB-iCreERT2

(*Claxton et al., 2008*), Ctnnb1$^{Ex3fl/fl}$ (*Harada et al., 1999*), BAT-gal$^{+/wt}$ Wnt/β-catenin reporter (*Maretto et al., 2003*) and *mT/mG* (*Muzumdar et al., 2007*).

Zebrafish (*Danio rerio*) were maintained under standard conditions at 28°C and a 14 hr light/10 hr dark cycle, in accordance with European and national animal welfare and ethical guidelines (protocol approval number: CEBEA-IBMM2017-22:65). Transgenic lines used in this study were Tg(kdrl:Hsa. HRAS-mCherry)$^{s896}$ (*Chi et al., 2008*) and Tg(7xTCF-Xla.Siam:GFP)$^{ia4}$ (*Moro et al., 2012*). After euthanasia with 0.3 mg.ml-1 Tricaine methanesulfonate (MS-222) for 10 min, adult brains aged 6 to 12 months were dissected and fixed overnight in sweet fixative (4% PFA, 4% sucrose in PBS). Brains were washed in PBS and embedded in 4% low-melting agarose. 300 µm sections were obtained using a LeicaVT1200s automated vibratome (Leica Biosystems). Sections were imaged on a Zeiss LSM710 confocal microscope using separate channels.

## Development assays

To investigate β-catenin activity in ECs, Wnt/β-catenin reporter mice (BAT-gal$^{+/wt}$) were bred with C57BL6/J mice to generate either heterozygous positive pups for β-galactosidase or homozygous negative control littermates. At different developmental stages (embryonic days E13.5, E17.5) embryos were harvested. For postnatal day 0 (P0) pups were sacrificed by decapitation, for postnatal day 21 (P21) pups were sacrificed by cervical dislocation. Brain preparation was performed in ice cold PBS and followed by overnight fixation in 4% PFA in PBS. For cryo-sectioning the whole brain was embedded in Tissue-Tek O.C.T. after incubation in 12/15/18% sucrose.

## Tightening of SFO vessels

β-Catenin endothelial specific gain of function system was kept by the use of Ctnnb1$^{Ex3fl/fl}$ (*Harada et al., 1999*) mice crossed with the Cdh5(PAC)-CreERT2 (*Wang et al., 2010*). To activate the Cre-recombinase, tamoxifen (TAM, 500 µg/day in corn oil; central pharmacy, Steinbach, Germany) was i.p. injected on five consecutive days. Brains were harvested and embedded for cryo-sectioning at 16, 19 and 26 days after the first TAM injection. To investigate the tightening effect at different postnatal stages, pups were i.p. injected with TAM (50 µg/day in corn oil) at P0-P3 and analyzed on P6 and P14.

## Tracer experiments

Animals were injected with TAM and kept for 26 days to assure SFO vessel tightening as described above. Mice were anesthetized and intravenously injected with 50 µl FITC-albumin (#A9771; Sigma-Aldrich). After 1.5 hr mice were sacrificed by cervical dislocation. The embedded brain was cryo sectioned (20 µm, counterstained for Podxl and analyzed after confocal imaging. For analysis the FITC covered area as well as the Podxl$^+$ vessel area within the SFO was measured for each optical section of at least one stack. To quantify the tracer leakage, the FITC covered SFO area was normalized to the vessel area, indicated by Podxl staining.

## Thirst inducing experiments

### Water restriction

To induce thirst, animals were water restricted for 72 hr. Therefore, the animals had no free access to water and got only a restricted amount of water every 24 hr according to their initial body weight

**Table 1.** Documentation of water provided to mice according to their body weight in the water restriction paradigm

| Bodyweight (BW) | Offered water [ml] |
| --- | --- |
| BW > 84% | 1.1 |
| 84% > BW > 83% | 1.2 |
| 83% > BW > 82% | 1.3 |
| 82% > BW > 81% | 1.4 |
| 81% > BW | 1.5 |

DOI: https://doi.org/10.7554/eLife.43818.034

(*Table 1*). During the experimental period the body weight is stable between at least 80–85% of the initial weight at day 0. After 72 hr there was no more water provided to keep the thirsty state. Mice were sacrificed and SFO tissue analyzed for c-fos as an immediate early gene marker for neuronal activity. The neuronal identity of c-fos[+] cells was confirmed by fluorescent Nissl co-staining (Key resource table).

## Hyperosmolar NaCl injection

Mice were i.p. injected with either 3 M or an isotonic (0.15 M) NaCl solution (150 µl/20 g mouse) as described in Zimmerman et al. (*Zimmerman et al., 2016*). After an incubation time of 50 min without any access to drinking water animals were sacrificed and SFOs were analyzed in cryo sections for c-fos activation.

## SFO whole mount for light sheet microscopy

### Sample preparation

To label blood vessels, 80 µl tomato lectin Alexa 649 (#DL-1178, Vector laboratories) were injected i.v. in adult mice. After 4 min of circulation time animals were sacrificed by cervical dislocation. After overnight fixation (4% PFA in PBS) whole mount tissue samples were blocked and permeabilized with 0.2% gelatin from bovine skin, Type B, 0.5% Triton in PBS from 24 hr up to one week according to their size. Antibodies (Key resource table) were applied in blocking buffer supplemented with 0.1% saponin. To fix the staining samples were incubated for one hour in 4% PFA.

## Tissue dehydration and clearing

At first the tissue was embedded in low melt agarose. The following dehydration and delipidation protocol was adapted from Orlich et al. and Renier et al. (*Orlich and Kiefer, 2018*; *Renier et al., 2014*). In brief, MetOH (50/70/100%) in PBS was used for 1 hrs for each step in dark-brown glass vials slightly shaking at RT, followed by an overnight incubation in 100% MetOH. To remove lipids an incubation with dichlormethane followed until the tissue sank down. Afterwards ethylcinnamate (ECi) (#112372, Sigma-aldrich) clearing was performed as described in Klingberg et al. (*Klingberg et al., 2017*). Samples were stored in ECi solution that was renewed one day before the acquisition. Samples were imaged in ECi solution with an UltraMicroscope II (LaVision, Germany) and stacks with 1 µm step size were further processed for visualization either by Imaris 9 (BitPlane, Switzerland) or the volume visualization framework Voreen (volume rendering engine) (*Meyer-Spradow et al., 2009*).

## Immunohistochemical staining

Either native frozen tissue or sucrose embedded samples were cryo-sectioned coronal or sagittal in 10 µm thickness and then fixed with 4% PFA for 10 min at room temperature or with ice cold MetOH for 3 min. To block/permeabilize tissue slides were incubated for 1 hr (overnight for vibratome section) (10% NDS, 0.1% Triton-X100 in PBS). Primary antibodies (Key resource table) were incubated for 2 hr (24 hr for vibratome section) and secondary for 1 hr (4 hr for vibratome section) in antibody incubation buffer (1% BSA, 0.1% Triton-X100 in PBS). If required, sections of PFA-fixed samples were subjected to antigen-retrieval (*) by boiling slides for 45 min in AR6 Buffer (#AR600250ML; Perkin Elmer). After cooling them down and an additional washing step, slides were stained as described above.

## Image acquisition and analysis of cryo-sections

Images were acquired using either a Nikon 80i wide field fluorescent microscope, or a Nikon C1si Confocal Laser Scanning Microscope, together with NIS-Elements Microscope Imaging Software for image analysis (Nikon Instruments, Inc., Düsseldorf, Germany). SFO vessels were defined as regions of interest (ROI) for area measurements. Staining was evaluated as a ratio of Cldn5 or Plvap/Meca32 to vessel area, evidenced by Podxl or Cdh5 labeling.

The number of c-fos[+] neurons as well as the total number of nuclei within the SFO, defined as ROI, were counted and the ratio of c-fos + to total nuclei was calculated.

**Table 2.** List of primers used for real time PCR.

| Primer for | Sequence 5'–3' sense | Sequence 5'–3' antisense |
| --- | --- | --- |
| qmm_Cldn5 | TGTCGTGCGTGGTGCAGAGT | TGCTACCCGTGCCTTAACTGG |
| qmm_Meca32 | CTTCATCGCCGCTATCATCCT | CCTTGGAGCACACTGCCTTCT |
| qmm_Rplp0 | GTGTTTGACAACGGCAGCATT | TCTCCACAGACAATGCCAGGA |
| qmm_Ocln | GTGAATGGCAAGCGATCATACC | TGCCTGAAGTCATCCACACTCA |

DOI: https://doi.org/10.7554/eLife.43818.035

## Electron microscopy

Animals were anesthetized and transcardially perfused with PBS/heparin for 1 min followed by 4 min with 4% PFA in cacodylate buffer (CB, pH 7.4). The SFO was whole mount prepared in ice cold PBS directly after brain isolation. Afterwards the tiny SFO whole mount tissue pieces were post-fixed with 4% PFA and 2% glutaraldehyde/CB overnight at 4°C.

Prior to embedding the tissue was incubated in 1% Os for 2 hr at RT followed by dehydration in graded acetone including contrast enhancement with uranyl acetate solution at 4°C o/n. Samples were embedded in Epon finally polymerized at 60°C for 24 hr. Ultra-thin sections (50 nm) were cut with Leica Ultracut UCT and analyzed using a Tecnai Spirit BioTWIN FEI electron microscope at 120kV. Images were taken with an Eagle 4K CCD bottom-mount camera. For the quantification of fenestrations, 5 SFO vessels were analyzed for each animal (n = 4 per genotype).

## RNA isolation, transcription and real time PCR analysis

RNA isolation was done using the RNeasy plus Microkit (Qiagen) according to the manufacturer recommendations with DNase on-column digestion (Qiagen) like suggested in the RNeasy Minikit (Quiagen). For cDNA synthesis (RevertAidTM H minus first strand cDNA synthesis kit, #K1632, Thermo Fisher Scientific) 57 ng RNA were used from SFO tissue of β-catenin GOF (Cre$^+$) and control (Cre$^-$) mice.

Quantitative real time RT-PCR (qRT-PCR) was performed in technical triplicates for each sample using the Absolute qPCR SYBR Green Fluorescein Mix (AB-1219, Thermo Fisher Scientific) according to the manufacturer's protocol. Rplp0 was used as a housekeeping gene for normalization. Expression data were analyzed with ΔΔct method. Primer sequences used for cDNA amplification by qRT-PCR are listed in *Table 2*.

## Statistical analyses

No statistical tests were used to predetermine sample size. Several independent experiments were performed to ensure reproducibility. The investigators were blinded by the experimental design during the analysis of the experiments shown in *Figures 4E*, *5F–H*, *6D*, *7E–F* and *8C* as well as in *Figure 4—figure supplement 1C*, *Figure 5—figure supplement 1D*, *Figure 7—figure supplement 2B*. Raw data are presented in the additional source data files.

The number of biological replicates is provided as 'n' in the legend of each figure. Technical replicates, such as the number of sections analyzed or replicates for qRT-PCR analyses are indicated in the figure legend and the respective material and methods section, respectively. Results are shown as mean ±SEM. Statistical significance was assessed by an unpaired t-test using GraphPad Prism version 6.0 (GraphPad Software Inc., USA). p-Values were considered significant at $p < 0.05$ and individual p-values are provided in each figure.

## Acknowledgements

This study was supported by the Deutsche Forschungsgemeinschaft SFB/TR23 'Vascular Differentiation and Remodeling', the research group FOR2325 'The Neurovascular Interface', the Excellence Cluster Cardio-Pulmonary System, the European Union HORIZON 2020 ITN 'BtRAIN', the German Centre for Heart and Circulation Research (DZHK, Column B: Shared Expertise), Landes-Offensive zur Entwicklung Wissenschaftlich-ökonomischer Exzellenz (LOEWE), Program of the Center for Personalized Translational Epilepsy Research, (CePTER) to SL. By the Goethe International Postdoc

Program, Go-In (291776) and the Goethe-University Junior Researchers in Focus Line A to SG. The Pdgfb-iCreERT2 mice were kindly provided by Marcus Fruttiger (University College London, Institute of Ophthalmology, London, UK), and the BAT-gal reporter mice were kindly provided by Stefano Piccolo (University of Padova, Dept. Molecular Medicine, Padova,Italy).

We thank Jochen Roeper, Natascha Diamantopoulou (Institute of Neurophysiology, Goethe University Frankfurt, Theodor-Stern-Kai 7, 60590 Frankfurt, Germany) for valuable advice regarding water restriction experiments and analysis of c-fos$^+$ Nissl staining. We further thank David Antonetti (Department of Molecular and Integrative Physiology, Kellogg Eye Center, University of Michigan, USA) and Andreas Mack (Institute for Clinical Anatomy and Cell Analytics, University of Tuebingen, Germany) for suggestions on antibodies for Ocln and Kir4.1, respectively. Moreover, we thank Sonja Thom for managing the mouse colonies, as well as Kavi Devraj, Burak Hasan Yalcin, Quinn Painter and Gabriela Hengel for excellent technical support and help.

## Additional information

### Funding

| Funder | Grant reference number | Author |
|---|---|---|
| Deutsche Forschungsgemeinschaft | LI 911/5-1 | Fabienne Benz<br>Ralf H Adams<br>Sylvaine Guérit<br>Stefan Liebner |
| Horizon 2020 Framework Programme | BtRAIN | Raoul FV Germano<br>Benoit Vanhollebeke<br>Stefan Liebner |
| Goethe University Frankfurt - Line A | | Sylvaine Guérit |
| Landes-Offensive zur Entwicklung Wissenschaftlich- ökonomischer Exzellenz (LOEWE), Program of the Center for Personalized Translational Epilepsy Research, CePTER | TP8 | Stefan Liebner |
| German Centre for Heart and Circulation Research (DZHK) | Column B: Shared Expertise | Stefan Liebner |

The funders had no role in study design, data collection and interpretation, or the decision to submit the work for publication.

### Author contributions

Fabienne Benz, Formal analysis, Investigation, Visualization, Methodology, Writing—original draft, Substantial contributions to conception and design, acquisition of data, or analysis and interpretation of data, Drafting the article or revising it critically for important intellectual content, Final approval of the version to be published; Viraya Wichitnaowarat, Martin Lehmann, Raoul FV Germano, Investigation, Visualization, Substantial contributions to acquisition, analysis and interpretation of data, Final approval of the version to be published; Diana Mihova, Investigation, Visualization, Substantial contributions to acquisition and analysis of data, Final approval of the version to be published; Jadranka Macas, Visualization, Substantial contributions to acquisition, analysis and interpretation of data, Final approval of the version to be published; Ralf H Adams, Sylvaine Guérit, Resources, Funding acquisition, Final approval of the version to be published; M Mark Taketo, Resources, Final approval of the version to be published; Karl-Heinz Plate, Conceptualization, Supervision, Investigation, Methodology, Substantial contributions to acquisition, analysis and interpretation of data, Final approval of the version to be published; Benoit Vanhollebeke, Investigation, Methodology, Substantial contributions to acquisition, analysis and interpretation of data, Final approval of the version to be published; Stefan Liebner, Conceptualization, Data curation, Software, Supervision, Funding acquisition, Visualization, Writing—original draft, Project administration,

Writing—review and editing, Substantial contributions to acquisition, analysis and interpretation of data, Final approval of the version to be published

**Author ORCIDs**
Fabienne Benz ⓘ http://orcid.org/0000-0002-1891-7680
Raoul FV Germano ⓘ https://orcid.org/0000-0002-1247-0689
Jadranka Macas ⓘ http://orcid.org/0000-0002-1661-2525
Ralf H Adams ⓘ https://orcid.org/0000-0003-3031-7677
Benoit Vanhollebeke ⓘ https://orcid.org/0000-0002-0353-365X
Stefan Liebner ⓘ http://orcid.org/0000-0002-4656-2258

**Ethics**
Animal experimentation: Animals were housed under standard conditions and fed ad libitum. All experimental protocols, handling and use of mice were approved by the Regierungspräsidium Darmstadt, Germany (FK/1052 and FK/1108). All animal handling was performed to minimize suffering.

**Decision letter and Author response**
Decision letter https://doi.org/10.7554/eLife.43818.043
Author response https://doi.org/10.7554/eLife.43818.044

# Additional files

### Supplementary files
• Transparent reporting form
DOI: https://doi.org/10.7554/eLife.43818.036

### Data availability
All data generated or analysed during this study are included in the manuscript and supporting files. Source data files are provided for Figures 4F-H, 5C, 6B, 7D, 8E-F, 9C as well as in Figure 4-figure supplement 1C, Figure 4-figure supplement 3D, Figure 8-figure supplement 1B. Raw data for all quantifications are provided in a separated MS Excel documents.

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
