## [Decision Letter]

Thank you for submitting your article "Low Wnt/β-catenin signaling determines leaky vessels in the subfornical organ and affects water homeostasis in mice." for consideration by *eLife*. Your article has been reviewed by three peer reviewers, including Elisabetta Dejana as the Reviewing Editor and Reviewer #1, and the evaluation has been overseen by K VijayRaghavan as the Senior Editor. The following individuals involved in the review of your submission have agreed to reveal their identity: Dritan Agalliu (Reviewer #2); Christer Betsholtz (Reviewer #3).

The reviewers have discussed the reviews with one another and the Reviewing Editor has drafted this decision to help you prepare a revised submission. The reviewers have found your paper of interest and potentially acceptable for publication in *eLife*. However, all the three referees raise important points to be considered before acceptance.

Summary:

The paper from Liebner laboratory investigates how CVO vessels, known to lack BBB and be permeable to water, solutes, and hormones, are regulated in relation to Wnt/β-catenin signaling – a central inducer of the BBB. The authors find (1) using Wnt reporter technologies, that Wnt signaling is undetectable in CVOs throughout development in mice and in adult zebrafish. (2) They describe a cellular phenotypic heterogeneity in the CVO vessels regarding Plvap and Cldn5 expression and they suggest that the latter might be regulated in a circadian fashion. (3) Using an inducible genetic β-catenin GOF approach, they show conversion of the CVO vessel phenotype into a BBB-like phenotype. (4) They show results that may indicate that this conversion has consequences for neuronal activity in the CVO.

This manuscript is an interesting and valuable contribution to the field that deserves publication in *eLife*. The manuscript is well-written and well-illustrated in general.

While going through the comments, please note in particular:

Additional controls for the role of canonical Wnt/β-catenin signaling in the observed control of permeability are asked for. If experimentally feasible, an earlier control of β-catenin signaling in SFO would also be important. Please see if these are speedily do-able and if not pleased adequately discuss these points. It will be useful to have this experimentally addressed, though.

A more accurate analysis of the permeability properties of SFO vessels and of tight junction organization and more accurate analysis of astrocyte polarization and matrix protein organization are also asked. More specifically, the authors should make better images for their c-fos data and show that it is indeed neurons that express c-fos.

The authors could remove the circadian Cldn5 data (premature), the Aqp4 and CD13 data (confusing).

Essential revisions:

The gain-of-function mutation of β-catenin is induced in endothelial cell specifically. It is known that endothelial cells can produce soluble neural mediators. An important question, therefore, is whether the observed neural hyperactivation in vivo, induced by β-catenin gain of function in endothelial cells, is due to reduced vascular permeability or to a change of endothelial cell angiocrine activity. This is a crucial aspect for a correct interpretation of the results. Would the authors be able to see the same c-fos upregulation in neural cells by limiting permeability with different tools? For instance, mice Meca32- PLVAP knock out? Src inhibitors? VEGF inhibitors? other more specific tools? If this these experiments are not speedily addressable, they should be carefully and frontally discussed.

A point that should be clarified is whether β-catenin does not play any role in the early angiogenesis of the SFO. The authors could not see, using reporter mice, a significant increase in β-catenin activity of the SFO-endothelial cells starting from 13.5E. However, β-catenin may be important in the very first steps of angiogenesis of SFO and then decline, as it occurs in other regions of the brain microcirculation. It could be, therefore, that it is already undetectable at E13.5 while active at earlier times. This is not irrelevant to understand SFO vascularization mechanisms and function. Here too, if experiments to address this are not speedily feasible, please discuss this point adequately.

A major functional feature of the BBB is to exclude various circulating molecules from entering the CNS. However, a functional assessment of barrier properties in SFO blood vessels after β-catenin stabilization is missing. The authors need to inject intravenously small or large MW tracers and assess their permeability into the SFOs under conditions of Wnt/β-catenin activation in endothelial cells.

Does overexpression of a stabilized form of β-catenin in the adult mice induce expression of Claudin-5 and suppresses MEca32 at the SFO?

Do the blood vessels in SFO acquire Occludin protein expression at cell junctions with β-catenin overexpression? The authors need to show some staining with Occludin antibodies.

The TEM data, clearly show that in GFO mice, pericytes and astrocytes are in close contact with endothelial cells. However, astrocytes do not appear to be polarized (AqP4 is expressed all over the parenchyma, rather than restricted to blood vessels). Are other astrocyte polarity markers (e.g. Dystroglycan or Kir 4.1) absent in this region?

One potential explanation for the observation that astrocytes do not form proper endfeet in the SFO region could be due to the absence of basement membrane proteins. The authors need to examine the expression of some of the key basement membrane proteins such as Collagen IV, Laminin α2 etc.

The authors nicely demonstrate that β-catenin GOF mice have higher c-Fos levels with water restriction than wild-type mice. Underwater restrictions, mice may show abnormal behaviors under specific behavioral tasks (e.g. Goltstein et al., 2018). It may be useful to test if there are any potential behaviors (e.g. rewarding behaviors) where GOF mice may perform worse than wild-type mice due to higher c-Fos activation.

The evidence for circadian regulation of Cldn5 expression weak. While the quantifications in Figure 4E may appear convincing, the immunostainings shown in Figure 4A-D are not convincing. The amount of vasculature captured in each section appears hugely variable and the Podxl staining for endothelium seems highly variable in strength. Podxl should be luminal and Cldn5 junctional (or is it?) and this is not at all apparent at the magnification shown. Also, if true, the circadian variation in Cldn5 immunoreactivity raises a number of important issues that should be addressed, including if it is junctional or non-junctional Cldn5 that varies, if the effect is transcriptional or not if other tight junction proteins such as occludin and ZO-1 are also regulated, and more. If it is true that TJs are regulated in a circadian fashion, it is obviously interesting, but the physiological consequences are unclear, given that Plvap expression is maintained, and therefore (presumably) fenestrations not regulated. Would TJ regulation matter for the permeability if fenestrations are kept? The authors speculate that the (possible) circadian regulation of Cldn5 could influence the circadian regulation of thirst and drinking behavior, but this is pure speculation, as far as I can judge. Again, how can Cldn5 matter on top of (presumably) open fenestrations? Given the many questions and the premature nature of this data, we suggest they be removed from the present manuscript, to allow a better focus on Wnt/β-catenin regulation of the CVO vessel phenotype. If Wnt signaling is involved in the circadian regulation of Cldn5 expression in CVO vessels, it would also make more sense to study short term effects of β-catenin GOF experiments; now the earliest time point studied in adults is 16 days following induction.

Another area where conclusions are weak relates to the TEM analysis (Figure 6). The images shown do not appear to support the conclusions that vessels in the GOF mice are slimmer, less tortuous and have less ECM deposition. For the ECM, it does appear denser in GOF but this may reflect decreased water content of the ECM rather than a decreased amount of ECM per se.

The data regards c-fos expression by neurons in the SFO are also an area where the conclusions do seem robust. Neuronal staining to show that this is neuronal and not glial localization is not there. Also, some of the staining patterns are puzzling. Should not c-fos should look similar at 0h and isotonic NaCl panels in Figure 7, but they look hugely different, casting doubt over the quantifications in Figure 7E and F.

The authors find increased expression of Sox17 along with Cldn5 in GOF experiments and note in conjunction with the TEM data that this may signify arterial specification. However, this is unfounded speculation in my opinion. There are many markers for arterial EC in the brain (see Vanlandewijck et al., 2018) and Sox17 is not a particularly strong one. The authors should remove this statement or investigate possible arterialization more thoroughly.

The authors investigate Aqp4 and CD13 staining patterns to check "other phenotypic BBB features", but although neither Aqp4 nor CD13 was found changed in the GOF situation, it is unclear why the experiments were done in the first place. In what way do the (negative) data display an absence of BBB features? The meaning of the homogenous-looking Aqp4 staining in the CVOs is interesting but may have nothing to do with the absence of a BBB. Does the staining reflect lack of end-foot restricted localization in astrocytes, or is it the tanycytes that express Aqp4 in the CVOs? Also, what does the CD13-staining mean? Is this mural cells or perivascular fibroblasts (see Vanlandewijck et al., 2018)?

---

## [Author Response]

Essential revisions:The gain-of-function mutation of β-catenin is induced in endothelial cell specifically. It is known that endothelial cells can produce soluble neural mediators. An important question, therefore, is whether the observed neural hyperactivation in vivo, induced by β-catenin gain of function in endothelial cells, is due to reduced vascular permeability or to a change of endothelial cell angiocrine activity. This is a crucial aspect for a correct interpretation of the results. Would the authors be able to see the same c-fos upregulation in neural cells by limiting permeability with different tools? For instance, mice Meca 32- PLVAP knock out? src inhibitors? VEGF inhibitors? other more specific tools? If this these experiments are not speedily addressable, they should be carefully and frontally discussed.

We thank the reviewer for getting into this discussion. Indeed, it is difficult, if not impossible to dissect the two effects downstream of β-catenin activation, namely the physical tightening of SFO vessels and the direct effects on endothelial cells, including their autocrine status.

**Author response image 1. respfig1:** 8 week-old female C57Bl6 mice were treated by i.p. injections of 1mg/kg BIO-X/DMSO every third day for 21 days. For the final 72 hours, mice were water-deprived and subsequently sacrificed and analyzed for Meca32 as well as for Cldn5 (n=10). Representative immunofluorescent images (**A**). Quantification of Meca32- and Cldn5-covered vessel area within the SFO (**B**). Error bars show ± SEM. Scale bars represent 100 µm.

Because of the time constraints of the revision period, we could not establish genetic mouse models for Meca-32/Plvap deletion in endothelial cells.

Although thoroughly addressing this question in vivo is a project on its own and would be beyond the scope of this manuscript, we have made an effort to repeat the water restriction experiments, using the Wnt pathway activator BIO-X (Author response image 1) as well as the Scr-inhibitor AZD0530 (Author response image 2).

We systemically administered BIO-X by i.p. injections of 1mg/kg BIO-X/DMSO every third day for 21 days. For the final 72 hours, mice were water-deprived and subsequently sacrificed and analyzed for Meca32 as well as for Cldn5. Unfortunately, we cloud not detect significant down regulation of Meca32 and up regulation of Cldn5 in SFO vessels. Instead, the phenotype of physiologically leaky vessels persisted.

Similar results were obtained, when mice were daily treated with 25mg/kg AZD0530 Scrinhibitor by oral gavage for 7 days. Scr inhibition did not result in any effects on Meca32 and Cldn5 expression evidenced by IF staining. Analysis of c-fos showed the expected increased staining in the SFO of water-restricted mice (data not shown).

**Author response image 2. respfig2:** 8 week-old female C57Bl6 mice were mice were daily treated with 25mg/kg AZD0530 Scr-inhibitor by oral gavage for 7 days. For the final 72 h, mice were water-deprived and subsequently sacrificed and analyzed for Meca32 as well as for Cldn5 (n=10). Representative immunofluorescent images (**A**). Quantification of Meca32- and Cldn5-covered vessel area within the SFO (**B**). Error bars show ± SEM. Scale bars represent 100 µm.

Currently, we are further analyzing the samples of the BIO-X- and AZD0530-treated mice for activation of β-catenin, by staining for nuclear β-catenin as well as for Lef-1, which will be informative for future experiments but will not be relevant for the revision of the present manuscript. Additional experiments with other compounds such as VEGF-inhibitors could not be performed due to time limitation.

To understand if activation of the Wnt/β-catenin pathway in ECs may alter their angiocrine profile, we have analyzed as a first step primary mouse brain microvascular endothelial cells (pMBMECs) that were stimulated with either Wnt3a or PBS/BSA control. Given the importance of vascular endothelial growth factor A (VEGFA) for neuronal function (“neurogenesis, neuronal migration, neuronal survival and axon guidance”, Mackenzie and Ruhrberg, 2012), we analysed VEGFA expression in Wnt-activated ECs by qRT-PCR. Despite robust induction of Axin2 as a well-established Wnt/β-catenin target, we did not observe a concomitant VEGFA regulation (Author response image 3).

**Author response image 3. respfig3:** pMBMECs were isolated and cultivated as previously published (Ziegler et al., 2016; Liebner et al., 2008), and after one passage stimulated for 9 days with 150mM recombinant mouse Wnt3a (#315-20, Peprotech). Total RNA was harvested (n=2), converted into cDNA and subjected to qRT-PCR analysis.

A point that should be clarified is whether β-catenin does not play any role in the early angiogenesis of the SFO. The authors could not see, using reporter mice, a significant increase in β-catenin activity of the SFO-endothelial cells starting from 13.5E. However, β-catenin may be important in the very first steps of angiogenesis of SFO and then decline, as it occurs in other regions of the brain microcirculation. It could be, therefore, that it is already undetectable at E13.5 while active at earlier times. This is not irrelevant to understand SFO vascularization mechanisms and function. Here too, if experiments to address this are not speedily feasible, please discuss this point adequately.

We thank the reviewer for this comment. We completely agree and have also analyzed earlier stages of BAT-gal reporter mice however, E13.5 was the earliest timepoint at which we could identify and visualize the developing SFO. Indeed, at this timepoint angiogenesis of the SFO is ongoing. Still, we could not observe active Wnt/β-catenin signaling. Consequently, due to the fact that E13.5 was the earliest timepoint the SFO is identifiable, we could not go earlier in order to study Wnt/β-catenin activation. Maybe we haven’t made this point as clear as it should be. We have now introduced this explanation in the Results section. We hope that this clarifies the reviewer’s question.

A major functional feature of the BBB is to exclude various circulating molecules from entering the CNS. However, a functional assessment of barrier properties in SFO blood vessels after β-catenin stabilization is missing. The authors need to inject intravenously small or large MW tracers and assess their permeability into the SFOs under conditions of Wnt/β-catenin activation in endothelial cells.

We thank the reviewer for this important comment. The physiological assessment of barrier tightness is indeed one of the most relevant parameters to address changes in permeability. In order to tackle the reviewer’s comment, we have injected β-catenin GOF and controls with tamoxifen, waited for 26 days and injected the mice intravenously with FITC-BSA (~68 kDa) and 3 kDa TMR-dextran. The detailed procedure is incorporated in the Material and Methods section of the revised manuscript. We now describe that the permeability for FITC-BSA was significantly reduced in the GOF compared to control animals. This result is now shown in Figure 6. Unfortunately, we encountered technical problems with a new Lot of the 3kDa TMR-dextran tracer that, although injected the same way as the FITC-BSA tracer, did not reveal the vessels nor was it detectable in the parenchyma of the SFO and other CVOs. For the reviewers’ reference, we show the positive technical pilot with an older Lot, nicely demonstrating leakage of the small tracer (Author response image 4). Unfortunately, the two experiments with the new Lot of the tracer failed and due to the restricted availability of double transgenic mice and of time for the revision we were not able to repeat this experiment.

We hope that the reviewers feel that we have sufficiently shown the physiological tightening of SFO vessels by the reduced leakiness of the ~68 kDa tracer and it clarifies the reviewers’ question.

**Author response image 4. respfig4:** 3 kDa TMR-dextran tracer was injected iv (150µl of a 2mM solution in PBS) into C56Bl6 WT mice for the pilot experiment (**A**), or iv into Ctrl and GOF mice in case of the real experiment (**B**). In the pilot experiment vessels were nicely filled and labeled by 3 kDa TMR-dextran (arrows). In the SFO (dashed line) tracer extravasation was visible. In the experiment with Ctrl and GOF mice, the tracer was not visible in the vessel lumen, although successful iv injection was confirmed by the larger tracer (FITC-BSA, main Figure 6).

Does overexpression of a stabilized form of β-catenin in the adult mice induce expression of Claudin-5 and suppresses MEca32 at the SFO?

We thank the reviewer for this question. It might be that we were not clear enough in the original submission of our manuscript, but the data that we’ve presented in the original Figure 5 (now Figure 4 in the revised manuscript) were derived from adult double transgenic mice. We have now clarified this in the figure legend as follows: “(A) Mouse model and (B) schedule of endothelial-specific β-catenin GOF induction by tamoxifen (TAM) in adult, 8-12 week old mice”.

Additionally, we have added data on the RNA expression of occludin (Ocln), showing in Figure 5 of the revised manuscript similarly to Cldn5 no mRNA upregulation could be observed, although Meca32 was significantly decreased in the SFOs of β-catenin GOF mice. Although the lack of transcriptional regulation of the two junctional genes may suggest that they are no direct targets of βcatenin, we might also face problems of signal masking by tight vessels in the total RNA, isolated from whole mount preparations. We now discuss this in the Discussion section and suggest that future RNAseq analysis of FACS-sorted pools and/or single cells may help to answer this question.

Do the blood vessels in SFO acquire Occludin protein expression at cell junctions with β-catenin overexpression? The authors need to show some staining with Occludin antibodies.

We thank the reviewer for this comment. We have investigated occludin (Ocln) expression and localization in SFO vessels of β-catenin GOF and control mice. The staining procedure and the antibody is incorporated in the Material and methods section of the revised manuscript. We now show that beside Cldn5, also Ocln localization at cell-cell junctions is significantly increased in GOF mice (Figure 5 of revised manuscript). Additionally, we have analyzed the expression of Ocln on the mRNA level by qRT-PCR and now show that in line with the expression of Cldn5 also the RNA of Ocln was not significantly increased in the SFO of GOF mice. As mentioned above, this might be due to the masking of the specific endothelial RNA by the total RNA isolated from the whole mount preparations, which inevitably include the ependymal epithelium that also expresses Ocln. Analyzing FACS-sorted cells from the SFO unfortunately, was beyond the scope of the present work.

In addition to Ocln, we have now also stained for the adherens and tight junction associated protein zonula occludens/tight junction protein 1 (ZO-/Tjp1), which we now show in Figure 5—figure supplement 1. As opposed to Cldn5 and Ocln, for ZO-1 we observed only a minor increase in junctional continuity of ZO-1 staining. This finding fits with a comparable distribution of VE-cadherin/Cdh5 in GOF and control vessels in the SFO (Figure suppl. 4), given that ZO-1 was shown to localize to adherens junctions if tight junctions are weak or absent (Tornavaca et al., 2015). This is now discussed in the Discussion section.

The TEM data, clearly show that in GFO mice, pericytes and astrocytes are in close contact with endothelial cells. However, astrocytes do not appear to be polarized (AqP4 is expressed all over the parenchyma, rather than restricted to blood vessels). Are other astrocyte polarity markers (e.g. Dystroglycan or Kir 4.1) absent in this region?

We thank the reviewer for bringing up this interesting issue. By showing the staining of Aqp4 for astrocytic endfeet and by CD13 for pericytes, our intention was to visualize potential changes to the entire neuro-vascular unit (NVU) in SFO vessels of β-catenin GOF compared to control mice. That the leaky vessels of the mouse SFO do not exhibit an organization as an NVU was previously shown by Pócsai et al., (2015). Indeed, we could not observe Aqp4 polarisation at astrocytic endfeet in the control conditions, but equally not along with barrier tightening in GOF mice. As suggested by the reviewer we now investigated the polarisation markers α-dystroglycan (α-Dag) and Kir4.1 and obtained comparable results now shown in Figure 7—figure supplement 1 of the revised manuscript, suggesting that end feet polarization does not follow endothelial tightening. Specifically, dystroglycan shows weak and not strictly polarized staining around SFO vessel, whereas Kir4.1 seems to be mainly expressed by cells morphologically resembling tanycytes in the SFO. The co-localized distribution of dystroglycan and Kir4.1 around BBB vessels nicely confirmed the specific staining of these markers. This was not only true 26 days after TAM injection but also after 60 day, supporting the interpretation that the proper cellular arrangement of the NVU requires additional cues than endothelial Wnt/β-catenin signaling.

One potential explanation for the observation that astrocytes do not form proper endfeet in the SFO region could be due to the absence of basement membrane proteins. The authors need to examine the expression of some of the key basement membrane proteins such as Collagen IV, Laminin α2 etc.

We thank the reviewer for this thoughtful comment. We have investigated collagen IV, laminin α2 (Lama2) with specific antibody staining in SFO vessels of β-catenin GOF and control mice. The staining procedure and the antibodies are incorporated in the Material and methods section of the revised manuscript. We now show that collagen IV and Lama2 are indeed expressed around SFO vessels, and their distribution and localization follows the outline of the vessels. At least from these observations we concluded that the lack of proper astrocytic endfoot formation around SFO vessels has no obvious correlation to a lack of the most abundant basement membrane components. These findings are shown in the revised manuscript in Figure 7—figure supplement 2.

The authors nicely demonstrate that β-catenin GOF mice have higher c-Fos levels with water restriction than wild-type mice. Underwater restrictions, mice may show abnormal behaviors under specific behavioral tasks (e.g. Goltstein et al., 2018). It may be useful to test if there are any potential behaviors (e.g. rewarding behaviors) where GOF mice may perform worse than wild-type mice due to higher c-Fos activation.

We thank the reviewer for this mindful comment. In the manuscript by Goldstein et al. the authors strictly compare food versus water restricted animals in their behaviour and conclude that there are little to no dramatic changes in overall behaviour due to water restriction. Specifically, water-deprived mice perform better regarding rewarding behaviour than food-deprived mice.

In order to tackle the reviewer’s request, we contacted our collaborators Prof. Jochen Roeper and Dr. Natascha Diamantopoulou (Institute of Neurophysiology, Goethe University Frankfurt), with which we have already planned to investigate the impact of SFO vessel tightening on mouse drinking behaviour. It turned out that it was not feasible to establish and conduct a meaningful experiment within the two month revision timeframe, considering 26d of TAM induction, mouse adaptation and taming, as well as water deprivation and analysis. Although testing the behaviour of Ctrl and GOF mice is beyond the scope of the present study, we believe however, that there is good evidence, that the overall rewarding behaviour does not play a major role in our paradigm. This has been shown and discussed in Augustine et al., 2018, who stated that activation of neurons in the median preoptic nucleus (MnPO), as part of the terminal lamina “indicate that MnPO^GLP1R^ neurons are activated purely by fluid consumption and not by reward-seeking behaviour or licking action per se. Consistent with the connection from MnPO^GLP1R^ to SFO^nNOS^ neurons, the activity of the SFO^nNOS^ population mirrored precisely the calcium dynamics of MnPO^GLP1R^ neurons”. The findings of Augustine et al. may therefore support our interpretation that, the increase in c-fos in the SFO of GOF water-deprived mice, relates to altered water homeostasis rather than altered overall behaviour. Moreover, we learned from our collaborators, that water deprivation as a widely used paradigm to augment the motivation of mice and other rodents, is not considered to lead to “abnormal” behaviour, given that water and food deprivation is the „normal “situation in wild mice, whereas water and food ad libitum could be rather considered as abnormal.

Nevertheless, we would like to emphasis that the question raised by the reviewer is very valid and that it was and is on our future agenda as it requires substantial expertise in behavioural sciences and adequate equipment (both will be provided by the group of Prof. Roeper). We hope the reviewers appreciate the limitations we faced regarding this question and leave the answer to future work.

The evidence for circadian regulation of Cldn5 expression weak. While the quantifications in Figure 4E may appear convincing, the immunostainings shown in Figure 4A-D are not convincing. The amount of vasculature captured in each section appears hugely variable and the Podxl staining for endothelium seems highly variable in strength. Podxl should be luminal and Cldn5 junctional (or is it?) and this is not at all apparent at the magnification shown. Also, if true, the circadian variation in Cldn5 immunoreactivity raises a number of important issues that should be addressed, including if it is junctional or non-junctional Cldn5 that varies, if the effect is transcriptional or not if other tight junction proteins such as occludin and ZO-1 are also regulated, and more. If it is true that TJs are regulated in a circadian fashion, it is obviously interesting, but the physiological consequences are unclear, given that Plvap expression is maintained, and therefore (presumably) fenestrations not regulated. Would TJ regulation matter for the permeability if fenestrations are kept? The authors speculate that the (possible) circadian regulation of Cldn5 could influence the circadian regulation of thirst and drinking behavior, but this is pure speculation, as far as I can judge. Again, how can Cldn5 matter on top of (presumably) open fenestrations? Given the many questions and the premature nature of this data, we suggest they be removed from the present manuscript, to allow a better focus on Wnt/β-catenin regulation of the CVO vessel phenotype. If Wnt signaling is involved in the circadian regulation of Cldn5 expression in CVO vessels, it would also make more sense to study short term effects of β-catenin GOF experiments; now the earliest time point studied in adults is 16 days following induction.

**Author response image 5. respfig5:** TEM images of SFO vessels taken from βcatenin control and GOF mice (26d after TAM injection) were analysed for vessels perimeter by measuring the perimeter of vessel using the polygon tool in ImageJ. In total 14 and 10 vessels were analysed for control and GOF, respectively. n=3; unpaired student t test was applied.

We thank the reviewers for this important comment. As suggested by the reviewers, we have removed the data on circadian regulation from the revised manuscript. Based on the other recommendations of the reviewers, we have now set Figure 5 of the original manuscript as Figure 4 and have included the novel data on occludin (Ocln) localisation and expression in SFO vessels of β-catenin GOF and control mice as Figure 5. We further thank the reviewer for the constructive suggestions on analyzing earlier time point after TAM induction, which we will consider in ongoing experiments

Another area where conclusions are weak relates to the TEM analysis (Figure 6). The images shown do not appear to support the conclusions that vessels in the GOF mice are slimmer, less tortuous and have less ECM deposition. For the ECM, it does appear denser in GOF but this may reflect decreased water content of the ECM rather than a decreased amount of ECM per se.

We thank the reviewer for this comment. As the description of vessels diameter and morphology in the TEM analysis was not based on any quantifications, we have analysed the vessel perimeter but could not identify a significant effect on vessel perimeter (Author response image 5). Consequently, we have removed the original note on vessel diameter and morphology from the revised manuscript. We also thank the reviewer for the comment on the ECM deposition. Indeed, it is impossible to judge if we face an increase of ECM in the GOF condition. We therefore have describe the vessels and the ECM in the revised manuscript as follows: “Although the vessel morphology did not show major differences regarding vessel perimeter and structure, the vessels of the GOF mice appeared to have a more compact ECM deposition in their circumference (Figure 7A-C).” (Results section). Together with the new Figure 7—figure supplement 2, showing the distribution of Lama2 and ColIV, we conclude that the ECM is not significantly affected by the induction of β-catenin signaling in ECs of the SFO.

The data regards c-fos expression by neurons in the SFO are also an area where the conclusions do seem robust. Neuronal staining to show that this is neuronal and not glial localization is not there. Also, I some of the staining patterns are puzzling. Should not c-fos should look similar at 0h and isotonic NaCl panels in Figure 7, but they look hugely different, casting doubt over the quantifications in Figure 7E and F.

We thank the reviewer for this important comment. We agree that the c-fos staining of the 0 hour water restricted mice and the NaCl controls should be comparable, given that we identified nearly no c-fos staining in these control conditions, which the images of the original submission did not support well, due to some background staining in the 0 h panel in Figure 7C. Moreover, the neuronal identity of cell that show nuclear c-fos staining was missing in the originally submitted manuscript. Therefore, we have performed novel staining of samples from the water-restricted mice, combining the vascular marker CD31 with the neuron-specific Nissl staining and c-fos. The novel images of the new Figure 8C, D now show a comparable staining pattern between water restriction at 0 hour and the isotonic NaCl injected mice. Additionally, the Nissl staining of the so-called Nissl flounders nicely documents the neuronal identity of the c-fos^+^ cells (Figure 8G). As opposed to the general nuclear staining by the fluorescent Nissl stain, the flounders are specific for neurons only, that we indicated by the dashed lines in Figure 8G. The images show a high magnification of a Nissl^+^/c-fos^+^ double positive neuron in Figure 8G and a Nissl^+^/c-fos^-^ neuron in Figure 8H. We hope the pronounced differences in c-fos between the conditions are now comprehendible.

The authors find increased expression of Sox17 along with Cldn5 in GOF experiments and note in conjunction with the TEM data that this may signify arterial specification. However, this is unfounded speculation in my opinion. There are many markers for arterial EC in the brain (see Vanlandewijck et al., 2018) and Sox17 is not a particularly strong one. The authors should remove this statement or investigate possible arterialization more thoroughly.

We thank the reviewer for this comment. The intention to show Sox17 was exclusively to demonstrate the upregulation of a Wnt/β-catenin target, and it was only a note that Sox17 was previously shown as an arterial marker. As suggested by the reviewer, we have deleted the statement of arterialisation, but described Sox17 solely as a Wnt target (Results section).

The authors investigate Aqp4 and CD13 staining patterns to check "other phenotypic BBB features", but although neither Aqp4 nor CD13 was found changed in the GOF situation, it is unclear why the experiments were done in the first place. In what way do the (negative) data display an absence of BBB features? The meaning of the homogenous-looking Aqp4 staining in the CVOs is interesting but may have nothing to do with the absence of a BBB. Does the staining reflect lack of end-foot restricted localization in astrocytes, or is it the tanycytes that express Aqp4 in the CVOs? Also, what does the CD13-staining mean? Is this mural cells or perivascular fibroblasts (see Vanlandewijck et al., 2018)?

We thank the reviewer for making this comment. As mentioned above in context of the astrocytic end foot staining with α-dystroglycan (α-Dag), Kir4.1 and Lama2, we decided to keep and show also Aqp4. However, we clarified in the main text of the revised manuscript (Results section; Discussion section) that the intention to investigate Aqp4, and as suggested by the reviewers in addition α-Dag and Kir4.1, was to understand if the known lack of a BBB-like NVU organisation of astrocytes, pericytes and endothelial cells in the core of the SFO, was altered by β-catenin GOF in ECs. The combined findings from all markers stained suggest that dominant endothelial activation of β-catenin does not lead to a structural formation of an NVU in the core of the SFO. We believe that this is valuable information, opening new aspects of investigating cellular and molecular interactions at the NVU. Regarding the expression of Aqp4 in astrocytes or tanycytes, the novel data on Kir4.1 expression which likely resembles tanycytes and does not overlap with Aqp4, may suggest that Aqp4 is expressed in astrocytes. We agree with the reviewer that the data on pericytes were too preliminary and hence, we have removed these data from the revised manuscript. Just mentioning that we haven’t seen obvious effects on this cell type as “data not shown”, and that we haven’t investigated perivascular fibroblasts identified by Vanlandewick et al., 2018 (Discussion section).